# Distinct G protein-coupled receptor phosphorylation motifs modulate arrestin affinity and activation and global conformation

Daniel Mayer[1,2,10], Fred F. Damberger [2], Mamidi Samarasimhareddy[3], Miki Feldmueller [1,2], Ziva Vuckovic[1,2], Tilman Flock [1,2,4], Brian Bauer[5], Eshita Mutt [1], Franziska Zosel [6], Frédéric H.T. Allain[2], Jörg Standfuss[1], Gebhard F.X. Schertler [1,2], Xavier Deupi [1,7], Martha E. Sommer[5], Mattan Hurevich [3], Assaf Friedler[3] & Dmitry B. Veprintsev [1,2,8,9]

Cellular functions of arrestins are determined in part by the pattern of phosphorylation on the G protein-coupled receptors (GPCRs) to which arrestins bind. Despite high-resolution structural data of arrestins bound to phosphorylated receptor C-termini, the functional role of each phosphorylation site remains obscure. Here, we employ a library of synthetic phosphopeptide analogues of the GPCR rhodopsin C-terminus and determine the ability of these peptides to bind and activate arrestins using a variety of biochemical and biophysical methods. We further characterize how these peptides modulate the conformation of arrestin-1 by nuclear magnetic resonance (NMR). Our results indicate different functional classes of phosphorylation sites: 'key sites' required for arrestin binding and activation, an 'inhibitory site' that abrogates arrestin binding, and 'modulator sites' that influence the global conformation of arrestin. These functional motifs allow a better understanding of how different GPCR phosphorylation patterns might control how arrestin functions in the cell.

[1] Laboratory of Biomolecular Research, Paul Scherrer Institute, 5232 Villigen, Switzerland. [2] Department of Biology, ETH Zürich, 8093 Zürich, Switzerland. [3] Institute of Chemistry, The Hebrew University of Jerusalem, Jerusalem, Israel. [4] Fitzwilliam College, Cambridge CB3 0DG, UK. [5] Institut für Medizinische Physik und Biophysik, Charité—Universitätsmedizin Berlin, Berlin 10117, Germany. [6] University of Zurich, Zurich 8057, Switzerland. [7] Condensed Matter Theory, Paul Scherrer Institute, 5232 Villigen, Switzerland. [8] Centre of Membrane Proteins and Receptors (COMPARE), University of Birmingham and University of Nottingham, Midlands NG7 2RD, UK. [9] Division of Physiology, Pharmacology & Neuroscience, School of Life Sciences, University of Nottingham, Nottingham NG7 2UH, UK. [10] Present address: Department of Pharmacology, University of California San Diego School of Medicine, La Jolla 92093-0636 California, USA. These authors contributed equally: Fred F. Damberger, Mamidi Samarasimhareddy. These authors jointly supervised this work: Fred F. Damberger, Martha E. Sommer, Mattan Hurevich and Assaf Friedler. Correspondence and requests for materials should be addressed to D.M. (email: damayer@ucsd.edu) or to D.B.V. (email: dmitry.veprintsev@nottingham.ac.uk)

G-protein-coupled receptors (GPCRs) detect and translate extracellular events such as changes in hormone or neurotransmitter concentration into intracellular responses by activating signaling effector proteins such as G proteins[1]. To control this signaling process, cells have developed a regulated system of GPCR desensitization beginning with their phosphorylation through specialized GPCR kinases (GRKs), which results in the subsequent recruitment and binding of arrestins[2,3]. Arrestins inhibit G-protein activation, mediate GPCR internalization, and possibly stimulate G-protein-independent signaling[4–9]. Phosphorylation of multiple sites within the C-terminus and/or intracellular loops of GPCRs is essential for the recruitment of arrestins[2,3,10]. The binding of arrestin-1 to rhodopsin has been reported to be controlled simply by the number of phosphorylated sites[10–12]. Other studies have suggested that different phosphorylation patterns on the intracellular C-terminal tail (the "phosphorylation barcode") of GPCRs can induce conformationally distinct active states of arrestins that result in a variety of cellular outcomes[13–24]. Recently, a common phosphorylation motif required for arrestin recruitment was proposed by Zhou et al. based on the crystal structure of a rhodopsin-arrestin-1 complex[25]. However, the proposed code is based on limited structural data (only two out of six potential phosphorylation sites were observed in the structure) and it does not account for the significant mass of published data indicating the functional importance of the other phosphorylation sites within the rhodopsin C-terminus[12,26–30]. The lack of consensus in the literature regarding the relative importance of the seven potential phosphorylation sites and the pattern of phosphorylation for the recruitment and activation of arrestin-1 motivated the current study.

Here, we systematically evaluate how the pattern of phosphorylation in the GPCR rhodopsin modulates affinity for arrestin, arrestin activation, and influences the global conformation of arrestin. Our approach was based on a library of synthetic phosphopeptides mimicking different phosphorylation states of the C-terminus of rhodopsin, and we measure arrestin affinity, activation and conformational modulation using a variety of biochemical and biophysical methods. Based on this analysis, we assign distinct functional roles to the individual phosphorylation sites, within a wider motif than described by Zhou et al.[25].

Our results help explain two outstanding questions in the field: (1) Why do some GPCRs interact transiently with arrestins while others form stable long-lived complexes, the so-called class "A" and class "B" receptors[31], and (2) Why do arrestin-2 and arrestin-3 have different preferences for these two receptor classes? The functional motifs we define here provide a molecular-level description of how GPCR phosphorylation patterns potentially control the cellular functions of arrestins[16,17] as well as a framework for interpreting the role of specific phosphorylation events in signaling outcomes.

## Results

**Phosphorylation sites that contribute to arrestin affinity.** Phosphorylation of at least two sites in the C-terminus of rhodopsin was reported to be necessary for arrestin recruitment, and three for arrestin activation[10]. Therefore, we first probed a peptide microarray of the rhodopsin C-terminus containing all possible combinatorial phosphorylation patterns from mono- to tri-phosphorylated peptides with a purified mCherry-arrestin-1 fusion protein (Fig. 1a). In total, there were 64 different peptides, including 7 with one, 21 with two and 35 with three phosphorylated serine and/or threonine amino acids. The data were analyzed with a linear regression model using feature selection to estimate the relative importance of individual phosphorylation

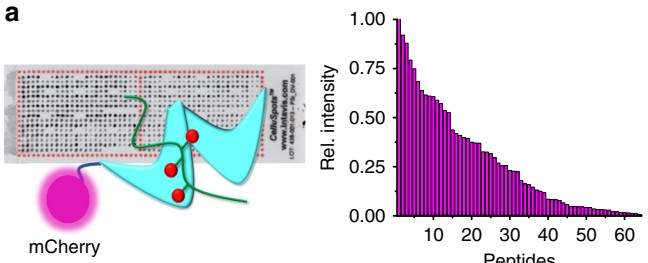

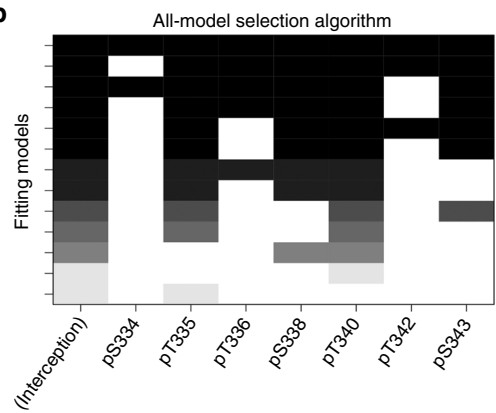

**Fig. 1** High-throughput screening of potential phosphorylation sites. **a** Schematic representation of the peptide array and the binding measurement using the mCherry-arrestin-1 fusion construct. Binding of the mCherry-Arrestin-1 fusion to each phosphopeptide was determined by the fluorescence at each spot on the array (barplot). **b** The fluorescence intensities of the acquired dataset were fitted to a linear model to determine the contribution of each position to arrestin binding. An all-subset model selection using the Akaike information criterion was used to identify the relative importance of each phosphorylation site to arrestin binding (see Methods). The plot shows the importance (fitting coefficients) of each phosphorylation site for the top 20 models (y-axis) in grayscale. The strength of the coefficient for a site in a model is indicated by the darkness of the boxes. White denotes that the coefficient was set to zero. The darker and longer columns indicate phosphorylation sites important for a tight interaction with arrestin-1. Interception represents the "non-specific" binding component

sites. The results suggested that phosphorylation of sites pT335, pS338, pT340, and pS343 had a significantly higher contribution to the interaction with arrestin-1 in comparison to pS334, pT336, and pT342 (Fig. 1b). Based on these findings, we designed a series of multiply phosphorylated peptides aimed at delineating the role of each site and the importance of the phosphorylation pattern in controlling arrestin affinity and activation (Table 1). The phosphopeptides were produced by a synthesis strategy that allowed us to synthesize peptides with several phosphorylation sites in very close proximity[32].

We next determined the affinities of these different phosphopeptides for arrestin-1 using fluorescence anisotropy. All peptides that showed tight binding ($K_d < 250\ \mu M$), namely peptides 7P, 6P, 4P, 3Pa, 3Pb, 3Pc, and 3Pd, contained pT340 and pS343 (Fig. 2, Table 2). Lack of phosphorylation at one or both of these sites resulted in a significant loss of affinity (peptides 5P, 3Pe, 3Pg, 3Ph, 3Pi, 3Pj, 3Pk), indicating that affinity depends not on the extent of phosphorylation but on the presence of pT340 and pS343, which we hereby term "key sites". Interestingly, a significant loss of arrestin-1 affinity was observed when pT342 was present in addition to the two key sites pT340 and pS343 (peptide 3Pf), suggesting the phosphate group at pT342 is an

**Table 1 List of all chemically synthesized peptides used in this study**

|  |  | pS334 | pT335 | pT336 | pS338 | pT340 | PT342 | pS343 |
|---|---|---|---|---|---|---|---|---|
| 7P |  | X | X | X | X | X | X | X |
| 6P |  |  | X | X | X | X | X | X |
| 5P |  | X | X | X | X | X |  |  |
| 4P |  | X |  | X |  | X |  | X |
| 3Pa |  | X |  |  |  | X |  | X |
| 3Pb |  |  | X |  |  | X |  | X |
| 3Pc |  |  |  |  | X | X |  | X |
| 3Pd |  |  |  | X |  | X |  | X |
| 3Pe |  |  |  | X | X |  |  | X |
| 3Pf |  |  |  |  |  | X | X | X |
| 3Pg |  |  | X |  | X | X |  |  |
| 3Ph |  |  | X | X |  |  |  | X |
| 3Pi |  |  | X |  | X |  |  | X |
| 3PJ |  |  | X | X | X |  |  |  |
| 3Pk |  |  |  | X | X | X |  |  |
| 0P |  |  |  |  |  |  |  |  |

ARGR-pT(347)-PP-pS(350)-LGPQDE-pS(357)-C-pT(359)-pT(360)-A-pS(362)-pS(363)-pS(364)-LAKDTSS
(V2Rpp)

The amino acid sequence of each peptide was identical (CDDEASTTVSKTETSQVAPA), and the peptides differed only regarding which serine and threonine residues were phosphorylated. The peptide sequence was derived from the bovine rhodopsin C-terminus with an N-terminal cysteine introduced to allow fluorescent labeling of the peptides. The fully phosphorylated C-terminus of V2 vasopressin receptor (V2Rpp, sequence indicated in black box) was included to have an additional reference in the trypsin digest assay and the fluorescence anisotropy experiments. For fluorescence anisotropy experiments, the V2Rpp was fluorescently labeled during the chemical synthesis at the N-terminus of the peptide

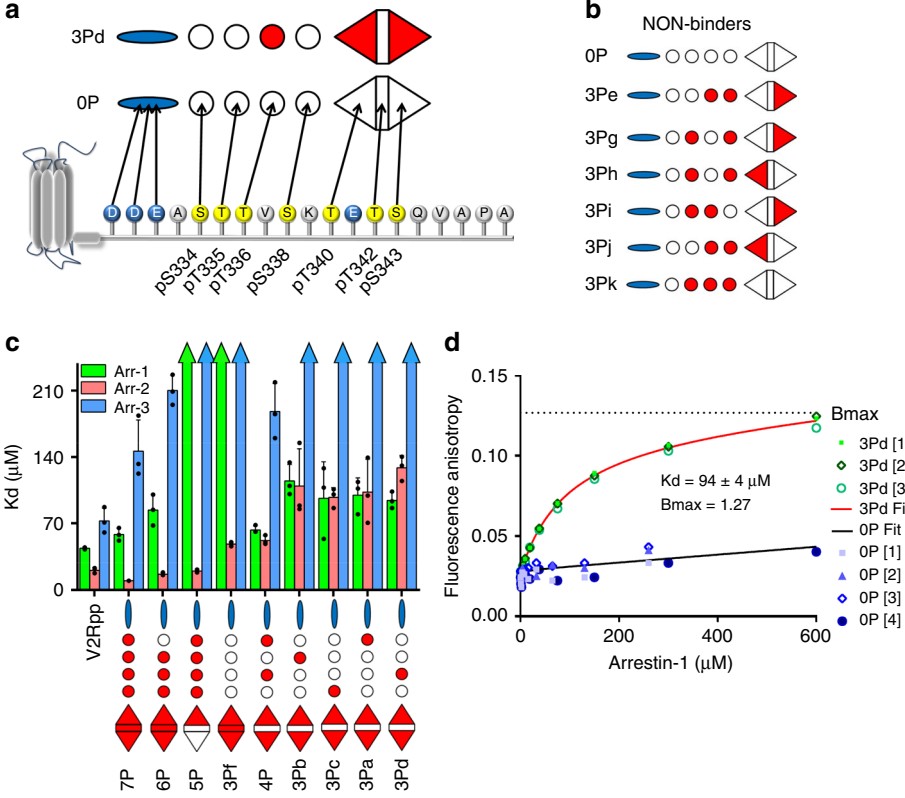

**Fig. 2** Different phosphorylation patterns have varying abilities to bind arrestins. **a** Sequence of the bovine rhodopsin C-terminus and schematic illustration of the phosphorylation and charge pattern. In the sequence, potential phosphorylation sites are yellow, and negatively charged acidic residues are blue. In the schematic, the negatively charged patch is blue, nonphosphorylated sites are white, and phosphorylated sites are red. The shapes of the phosphorylation sites indicate their different functions, which were assigned in this study. **b** Peptides that did not show specific binding or very weak binding to any tested arrestin isoform. **c** $K_d$ values of all specific binders determined by fluorescence anisotropy for arrestin-1, -2, and -3. Statistical analysis was performed by ANOVA analysis and these values are reported in Supplementary Information Tables 1, 2 and 3. **d** Fluorescence anisotropy arrestin-1 binding curves of nonphosphorylated peptide 0P (different shades of blue) and the tri-phosphorylated peptide 3Pd (different shades of green). Each titration was repeated three or four times. Fits of the individual binding curves were used to obtain error bars shown in part (**c**) and Table 2 (Source data are provided as a Source Data file.)

**Table 2 Summary of determined affinities and trypsin digestion results**

| | arrestin-1 $K_d$ /µM | | arrestin-2 $K_d$ /µM | | arrestin-3 $K_d$ /µM | | trypsin digest of arrestin- 1 | 2 | 3 |
|---|---|---|---|---|---|---|---|---|---|
| 7P | 58 ± 5 | | 21 ± 2 | | 146 ± 25 | | + | Faster | + |
| 6P | 84 ± 11 | | 17 ± 1 | | 210 ± 11 | | + | Faster | + |
| 5P | >250 | | 20 ± 1 | | >250 | | − | Normal | − |
| 4P | 63 ± 3 | | 52 ± 4 | | 201 ± 38 | | + | Faster | + |
| 3Pa | 100 ± 14 | | 103 ± 28 | | >250 | | + | Normal | + |
| 3Pb | 115 ± 13 | | 110 ± 30 | | >250 | | + | Normal | + |
| 3Pc | 97 ± 28 | | 98 ± 8 | | >250 | | + | Normal | + |
| 3Pd | 94 ± 6 | | 129 ± 9 | | >250 | | + | Normal | + |
| 3Pe | >250 | | >250 | | >250 | | − | Normal | − |
| 3Pf | >250 | | 48 ± 1 | | >250 | | − | Normal | − |
| 3Pg | >250 | | >250 | | >250 | | − | Normal | − |
| 3Ph | >250 | | >250 | | >250 | | − | Normal | − |
| 3Pi | >250 | | >250 | | >250 | | − | Normal | − |
| 3Pj | >250 | | >250 | | >250 | | − | Normal | − |
| 3Pk | >250 | | >250 | | >250 | | − | Normal | − |
| 0P | >250 | | >250 | | >250 | | − | Normal | − |
| V2Rpp | 44 ± 1 | | 12 ± 2 | | 73 ± 10 | | − | Faster | − |

Affinities of differently phosphorylated peptides to arrestin-1, -2 and -3 (see Fig. 2) are listed on the left side of the table, and the summary of limited trypsin digest results for arrestin-1, -2 and -3 (see Fig. 3 and Supplementary Figure 1) is listed on the right side of the table. All affinity measurements were performed in triplicate. The determined dissociation constants ($K_d$) are listed. The $K_d$ values of noninteracting or weakly interacting phosphopeptides that could not be determined because they were below the detectable range of the assay are indicated as ">250". Phosphopeptides that induced a change in the digest pattern of arrestin-1 and arrestin-3 from three to two bands are indicated by "+", and those that failed to induce a change are indicated by "−". In our hands, trypsin digestion of arrestin-2 always resulted in two bands, and the presence of certain phosphopeptides increased the rate of digestion. Increased rate of digestion is indicated by "Faster", and no change in rate is indicated by "Normal" (Source data are provided as a Source Data file.)

"inhibitory site" that can interfere with the interaction of pT340 and/or pS343 with arrestin-1.

We also evaluated the binding of the phosphopeptides to nonvisual arrestin-2 and -3, which also interact with phosphorylated rhodopsin[33]. Phosphopeptide binding affinity to arrestin-2 generally increased with the degree of phosphorylation, and in contrast to arrestin-1, arrestin-2 bound well to peptide 5P (lacking key site pS343) and 3Pf (containing inhibitory site pT342) (Fig. 2c, Table 2). This behavior can be explained by increased mobility and displacement of the C-terminal tail (C-tail) of arrestin-2 (see below). For arrestin-2 binding to tri-phosphorylated peptides, the key sites pT340 and pS343 were required for tight binding, similar to arrestin-1. Significantly weaker binding to arrestin-3 was observed for all phosphopeptides in comparison to arrestin-1 and arrestin-2, which is consistent with previous in vivo measurements[31,34]. Despite this weaker binding, arrestin-3 was similar to arrestin-1 regarding which phosphorylation patterns were preferred, showing higher affinity for 7P, 6P and 4P peptides (containing key sites pT340 and pS343) compared to 5P (lacking pS343), and abrogated binding when inhibitory site pT342 was present (Fig. 2c, Table 2). All tested arrestin isoforms showed high affinity for a phosphopeptide analog of the fully phosphorylated C-terminus of the V2 vasopressin receptor (V2Rpp) (Fig. 2c), a "class B" GPCR that has been reported to robustly interact with arrestin-1, -2 and -3 [31].

**Key sites are required for conformational change in arrestin**. Limited trypsin digestion has been used previously to detect conformational changes in arrestins that are induced by phosphorylated peptides derived from different GPCRs[23,24,35]. The digestion of arrestin-1 in the presence of nonphosphorylated peptide resulted in three major bands, whereas in the presence of the fully phosphorylated peptide 7P two major bands were observed (Fig. 3a). This change in the digest pattern has been

linked to C-tail displacement in arrestin-1 [36]. Importantly, only peptides that contained both the key sites pT340 and pS343 were able to influence the digest pattern of arrestin-1 with the notable exception of 3Pf containing the inhibitory site pT342 (Fig. 3, Table 2).

In the case of arrestin-2, the application of limited amounts of trypsin resulted in relatively fast and efficient digestion, and the pattern of digested protein bands was not significantly changed by the presence of phosphopeptide (Fig. 3a). This difference in protease sensitivity likely reflects an increased mobility and spontaneous displacement of the C-tail in arrestin-2. This increased C-tail flexibility allows arrestin-2 to bind phosphopeptides with higher affinity than arrestin-1 and arrestin-3 (Fig. 2c, Table 2), since C-tail displacement exposes the putative phosphopeptide binding site[37]. This mechanism underlies the significantly enhanced affinity of C-terminally truncated arrestin-1 (p44) for the phosphorylated C-terminus of rhodopsin[38,39]. Note that digestion of arrestin-2 was accelerated in the presence of 7P, 6P, and 4P peptides, but not in the presence of the other binding peptides, including 5P (Table 2, Supplementary Figure 1). Considering that all peptides containing phosphorylation at both key sites—as well as peptide 5P—bound arrestin-2 with similar high affinity (Table 2), these differences in trypsin sensitivity do not reflect different levels of phosphopeptide association with arrestin-2 but rather differences in the arrestin-2 structure induced by the different peptides.

Limited trypsin digestion of arrestin-3 yielded very similar results as for arrestin-1, including the requirement of the key sites pT340 and pS343 to stimulate C-tail displacement (Fig. 3a, Table 2). Hence, the key sites are important not only for high-affinity binding but also for inducing conformational changes in arrestin-1, -2 and -3.

**Key sites activate arrestin for receptor binding**. Since some phosphopeptides were able to displace the C-tail of arrestin, and

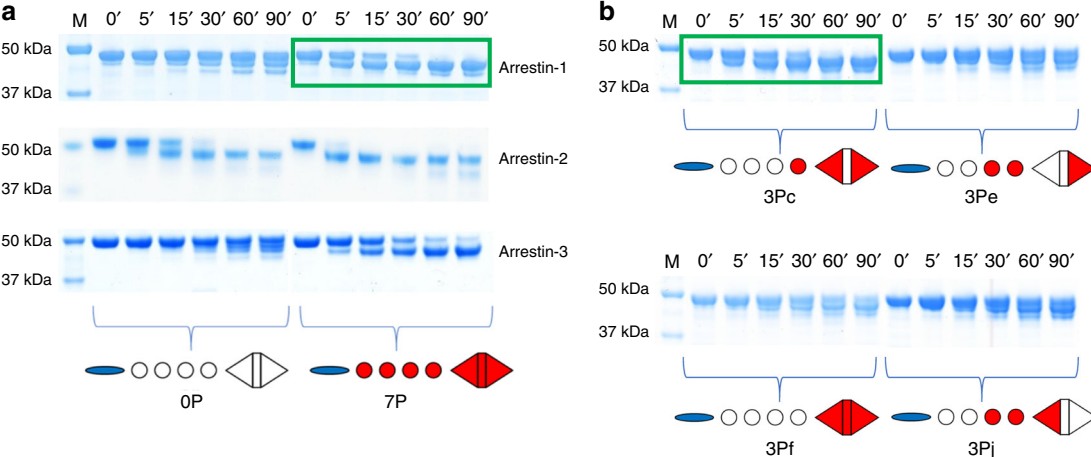

**Fig. 3** Conformational change of arrestins depend on phosphorylation pattern. **a** Limited trypsin digest of arrestin-1, -2 and -3 in presence of non- (0P) and fully phosphorylated (7P) phosphopeptide analogs of the rhodopsin C-terminus. The presence of fully phosphorylated peptide resulted in a change of the digestion pattern from three to two bands in comparison to nonphosphorylated peptide for arrestin-1 and -3. The digestion of arrestin-2 always resulted in two bands with similar molecular weight as arrestin-1/-3 independent of which peptide was present. However, digestion was accelerated in the presence of fully phosphorylated peptide as compared to nonphosphorylated peptide. **b** Limited trypsin digestion of arrestin-1 in the presence of different tri-phosphorylated peptides. Peptide-induced conformational changes in arrestin-1 are indicated by a change in the digest pattern (green boxes) (Source data are provided as a Source Data file.)

this is an essential step to activate arrestin for binding to the activated receptor, we reasoned that these phosphopeptides would facilitate binding of arrestin to activated nonphosphorylated receptor (as has previously been demonstrated using the fully phosphorylated C-terminal peptide derived from rhodopsin[40]). We therefore employed site-specific fluorescently labeled mutants of arrestin-1 to evaluate the potential of the different phosphopeptides to induce arrestin activation and receptor binding. Specifically, we quantified ternary complexes of fluorescently labeled arrestin-1 mutants, nonphosphorylated rhodopsin in native rod outer segment (ROS) membranes, and differently phosphorylated peptides (Fig. 4a). We measured fluorescence changes of arrestin-1 mutants I72B and F197NBD (incorporating bimane and NBD fluorescent labels respectively at positions I72 and F197; see Methods and Fig. 4b−d). I72B is located in the finger loop that engages the transmembrane core of rhodopsin[41,42]. F197NBD is located at the C-edge of the C-terminal domain that is known as the membrane anchor, a binding element that is distinct from the receptor[43]. These fluorescently labeled arrestins have been shown to report arrestin-1 engagement of the active receptor core and membrane, respectively[43,44]. Binding of the two fluorescently labeled arrestin-1 mutants was measured in parallel using a centrifugal pull-down assay (Supplementary Figure 2). The pull-down results mirrored the results obtained by fluorescence for both I72B and F197NBD arrestin mutants; hence, the observed fluorescence changes represent arrestin activation and tight binding to the active receptor. One exception was the ternary complex formed with the 7P peptide and arrestin-1 I72B, likely due to quenching of the bimane on the finger loop by electron-rich groups on the phosphopeptide. No quenching effects were observed for arrestin-1 F197NBD, in which the NBD fluorophore is located far from the putative phosphopeptide binding site[25,37]. We observed that different phosphopeptides had varying abilities to induce ternary complex formation, and the same trend was observed for both fluorescently labeled arrestin-1 mutants (Fig. 4e). Efficient ternary complex formation was detected for all peptides containing the two key sites pT340 and pS343 (7P, 6P, 4P, 3Pa, 3Pb, 3Pc, 3Pd) except for 3Pf with the inhibitory site pT342. Lack of one of these

two key sites, such as in peptide 5P, resulted in significant reduction of ternary complex formation. Tri-phosphorylated peptides clustered into three groups: those with both key sites that showed robust ternary complex formation (3Pa, 3Pb, 3Pc, 3Pd), others lacking one or both key sites that induced very little arrestin binding (3Pe, 3Ph, 3Pj, 3Pk), and two lacking one of the key sites but containing both pT335 and pS338 that showed moderate ternary complex formation (3Pg, 3Pi, marked with red arrows in Fig. 4e). Interestingly, these sites showed a higher contribution to arrestin-1 interaction than pS334, pT336 and pT342 in the peptide array experiment (Fig. 1b). Hence, we conclude that pT335 and pS338 serve as secondary sites that enhance arrestin affinity and activation in the absence of phosphorylation at one of the key sites. A noteworthy outlier in this grouping of tri-phosphorylated peptides is the 3Pf peptide, which stimulated less ternary complex formation because of the presence of the inhibitory site pT342, despite having both key sites pT340 and pS343.

Notably, the relative levels of ternary complex formation are highly consistent with the other data presented in this study. These data further show that the phosphopeptides that bind arrestin-1 with high affinity and stimulate C-tail release also activate arrestin for receptor core binding. This finding supports the recently described allosteric activation mechanism for arrestin by Latorraca et al.[45], where engagement of the binding site for the phosphorylated receptor C-terminus stabilizes an active conformation of arrestin that can couple to the active receptor core.

**Global conformation modulation of arrestin-1.** We used [15N,1H]TROSY NMR spectroscopy to characterize the location of conformational changes in arrestin-1 induced by differently phosphorylated phosphopeptides. Chemical shift changes of the detected amide signals report changes in the local environment of investigated amino acids, such as changes in polypeptide backbone geometry, H-bonding or polar interactions or changes in its position with respect to other amino that influences the magnetic field of the studied nuclei. A representative example of such an NMR experiment to monitor phosphopeptide 7P binding to arrestin-1 is shown

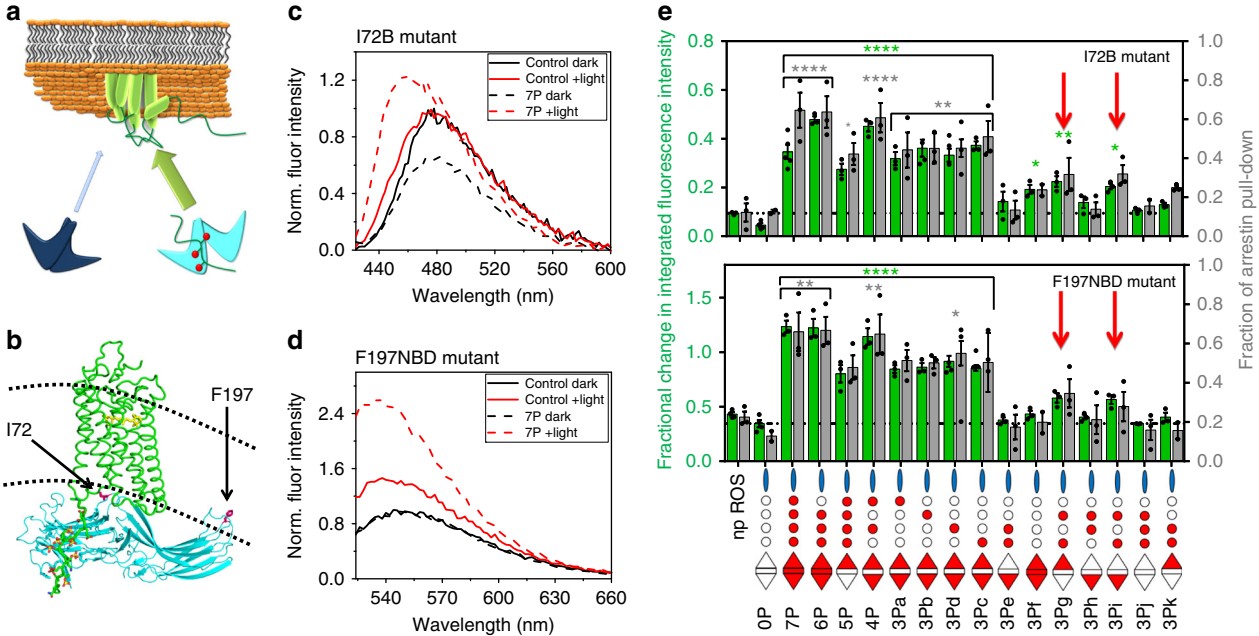

**Fig. 4** Ternary complex formation depends on peptide phosphorylation pattern. **a** Schematic representation of phosphopeptide-induced arrestin-1 activation and recruitment to nonphosphorylated light-activated rhodopsin in the native membrane. **b** Model of a complex of arrestin-1 bound to active phosphorylated rhodopsin (own work). Locations of the introduced fluorescent probes on the finger loop (I72) and C-edge (F197) are indicated, as is the membrane bilayer (dashed lines). **c** Steady-state fluorescence spectra of arrestin-1 mutant I72B in the presence of nonphosphorylated rhodopsin, measured in the dark-state (black spectra) and after light-activation (red spectra), in the absence (solid spectra) or presence (dashed spectra) of fully phosphorylated peptide 7P. **d** Steady-state fluorescence spectra of arrestin-1 mutant F197NBD, as described in (**c**). **e** Quantification of ternary complex formation using arrestin-1 mutants I72B and F197NBD as measured by fluorescence (green bars, left axes) and centrifugal pull-down analysis (gray bars, right axes). The results of a statistical analysis (ANOVA) of each peptide versus "np ROS" are indicated by green (fluorescence) and gray (pull-down) stars above each column (*$P \leq 0.05$, **$P \leq 0.01$ and ****$P \leq 0.0001$. The red arrows indicate tri-phosphorylated peptides which lack one of the key sites pT340 or pS343 but include the weak recruiter (secondary) sites pT335 and pS338. Data represent the average± s.e.m. from three independent experiments

in Supplementary Information Fig. 3. We based our assignments of the signals on a previous study[46] and used triple resonance experiments to confirm slight changes in the signal positions of some residues in our different buffer system. Each titration was performed in a stepwise manner up to a peptide:protein ratio of 10:1 (see spectra expansion in Fig. 5). Based on the measured $K_d$ values (Table 2), this final ratio corresponded to a peptide saturation of 88–93% for all samples except the 5P peptide which had a slightly lower occupancy of approximately 79%.

All titrated peptides resulted in numerous peaks with reduced intensity or shifted position with gradually decreasing intensity with the noteworthy exception of peptide 5P (Supplementary Figure 4). A number of peaks appeared at new positions suggesting the reappearance of intermediate exchange broadened signals at close to their final positions due to a change in either the populations or the timescale of averaging between multiple conformational states for the residues in question (Figs. 5, 6, Supplementary Figure 5, Supplementary Information Table 4). In some cases we detected multiple nearby peaks suggesting the presence of several states in slow conformational exchange on the NMR timescale (Fig. 5a, b), as reflected by the splitting of the peak at high concentration of the phosphorylated peptide. The majority of chemical shift changes were detected in the N-domain of arrestin-1, and the largest changes were observed at sites near the putative binding site of the phosphorylated receptor C-tail[25,37], including the so-called three-element interaction and polar core (interaction networks that stabilize the basal conformation of arrestin[47]) (Figs. 5, 6, 7, Supplementary Figures 3, 4 and 5, Supplementary Information Table 4). The chemical shift

changes of eight representative residues located in different regions of arrestin-1 are shown in Fig. 7. We observed that peptides containing the two key sites, pT340 and pS343, induced some of the largest chemical shift changes in the vicinity of the polar core (E39, L172, G389), suggesting that these phosphorylation sites disrupt the central stabilizing element of the basal arrestin conformation. The magnitude of the chemical shift changes of G389 was very similar for all strongly binding peptides, which likely reflects their common ability to induce C-tail displacement (Figs. 3, 7 and Table 2). The 5P peptide, lacking one of the key sites (pS343), caused minimal chemical shift changes, which is consistent with its inability to induce conformational changes in arrestin as measured by trypsin digestion (Table 2). Surprisingly, this weakly interacting peptide could still stimulate ternary complex formation (Fig. 4e). This ability may arise because the active receptor increases the apparent affinity of the 5P peptide for arrestin-1 by an allosteric activation of arrestin via the receptor−core interface[45].

Interestingly, the magnitude of chemical shift changes depended on which phosphopeptide was bound to arrestin-1, in particular for the finger loop (G76), the beta sandwich of the C-domain (L240 & E262), and the long inter-domain loop between β-strands 17 and 18 (I312). The chemical shift changes observed at sites far away from the putative phosphopeptide binding site might be due to inter-domain twisting that affects solvent accessibility or changes in H-bond length at the β-strand interfaces (Fig. 7). Inter-domain twisting is a hallmark of arrestin-1 activation[39,42] and has been observed in the crystal structure of arrestin-2 bound to a vasopressin V2 receptor-derived

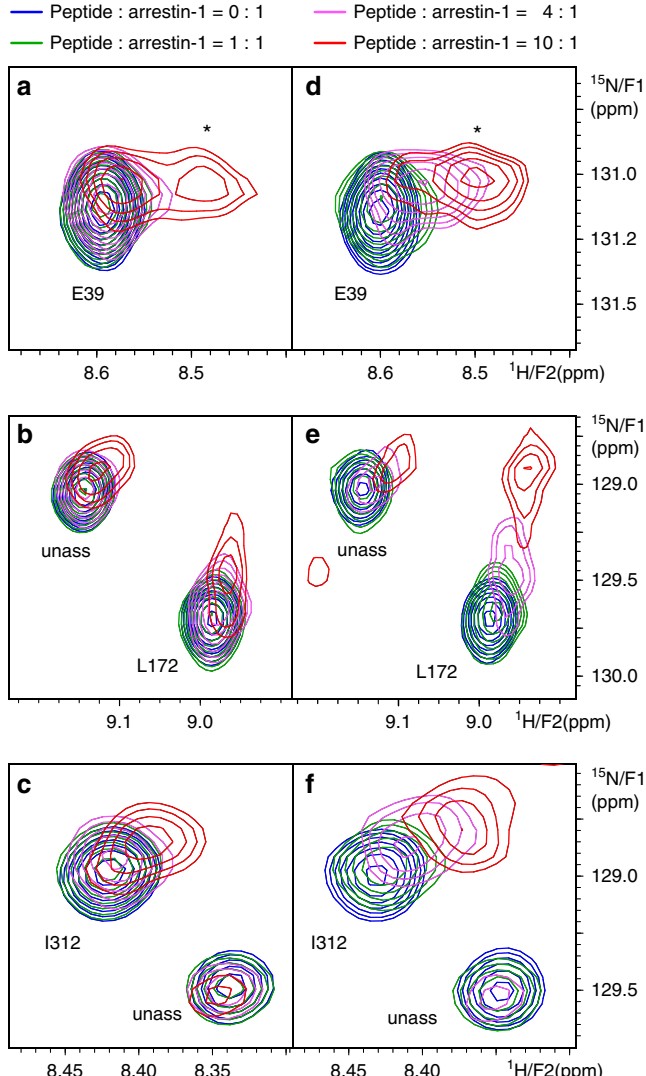

Legend:
Peptide : arrestin-1 = 0 : 1 (blue)
Peptide : arrestin-1 = 1 : 1 (green)
Peptide : arrestin-1 = 4 : 1 (violet)
Peptide : arrestin-1 = 10 : 1 (red)

**Fig. 5** Expansions of the series of [$^{15}$N,$^1$H]TROSY spectra of arrestin-1. It shows representative chemical shift changes in arrestin-1 induced by the 3Pa peptide (**a**−**c**) and the fully phosphorylated peptide 7P (**d**−**f**, see Supplementary Figure 3 for full spectrum) for E39, L172, and I312 at peptide: arrestin-1 ratios of 0:1 (blue), 1:1 (green), 4:1 (violet), and 10:1 (red). Unassigned resonances are labeled "unass". A second peak appearing at 10:1 in (**a**) and (**d**) which may indicate the presence of multiple states in slow exchange is marked with an asterisk

phosphopeptide[37]. These changes may also reflect the dynamic sampling of different conformational states which is consistent with the differential line broadening (as reflected by the reduction of the peak intensity) observed for sites distal to the peptide binding site during titration of strong binders (Fig. 5c, f). Significantly, these results suggest that the global conformation and dynamics of arrestin-1 depends on the phosphorylation pattern. Line broadening observed at the putative peptide binding interface (Figs. 5b, e, 6, 7, Supplementary Figure 3 and 5) can be interpreted as due to a mixture of several peptide-bound conformational states. This finding is anticipated given the large number of basic residues in the N-domain of arrestin that could accommodate different phosphopeptide binding modes and the flexibility of phosphopeptide-bound arrestin in the absence of the active receptor–core interface[17].

## Discussion

The principal findings of this study are summarized in Fig. 8a. The "key sites" we define here (pT340 and pS343 in rhodopsin) are essential for the formation of a high-affinity complex between arrestin and a GPCR. The crystal structure of arrestin-2 bound to V2Rpp[37] indicates how the key sites both perturb the polar core and interact with a critical phosphorylation sensor near the three-element interaction site[48,49]. These two interactions thereby stabilize the active conformation of arrestin-2 as illustrated in Fig. 8b. Upstream of the key sites, pS350 on V2Rpp, forms hydrogen bond contacts with the finger and middle loops of arrestin-2, which might activate them for receptor binding[41,50]. In rhodopsin, this structural requirement for arrestin-1 activation could be provided by a negatively charged patch consisting of D330 and D331, which we term the "negatively charged region". The four phosphorylation sites between the negatively charged region and the activator sites (pS334, pT335, pT336, and pT338) we designate "modulator sites" (Fig. 8a), since both the biochemical data and the NMR data show that the global conformation and flexibility of arrestin-1 depends on which of these sites are phosphorylated. To summarize, the GPCR phosphorylation motifs we identify here control not only the overall conformation of arrestin, but its activation state, its ability to couple to the active receptor core, and the relative stability and expected lifetime of the arrestin-GPCR complex. Such knowledge is ultimately necessary to understand the molecular mechanism by which different receptor phosphorylation patterns determine how arrestin interacts with different GPCRs and influences cellular functions of arrestin (e.g. receptor desensitization and endocytosis).

Two previous in situ studies have suggested that phosphorylated threonines may be more important for arrestin recruitment to rhodopsin than phosphorylated serines[12,29]. However, these studies cannot be directly compared to ours, since the exact levels of phosphorylation could not be controlled. Potential sites of phosphorylation were removed genetically, and phosphorylation of the remaining sites depended on the native rhodopsin kinase (GRK1) in the rod cells. Notably, rhodopsin containing only threonines was not effectively phosphorylated (>50% not phosphorylated at all), and its deactivation in the rod cell was apparently independent of GRK1 and arrestin-1 [29]. These results suggest that other, arrestin-1-independent mechanisms might exist in the rod cell to deactivate this mutant rhodopsin.

In addition we discovered that site T342, which lies between the two key sites, plays a complex inhibitory role. When this site is phosphorylated along with the two key sites, the affinity of the peptide for arrestin-1 and -3 is significantly reduced as compared to other tri-phosphorylated peptides with the key sites phosphorylated. However, phosphorylation of T342 did not affect arrestin-1 or -3 affinity in the case of more highly phosphorylated peptides (e.g. 7P, 6P and 4P). How might phosphorylation between the two key sites be either inhibitory or neutral, depending on the overall phosphorylation level? In the crystal structure of arrestin-2 bound to V2Rpp[37], the key phosphorylation sites pT360 and pS363 (analogous to pT340 and pS343 in rhodopsin) interact with multiple "phosphosensors" near the polar core and three-element interaction on arrestin-2, while the inhibitory site pS362 (analogous to pT342) is more solvent exposed (Fig. 8b). The phosphate group at site pS362 appears to form a hydrogen bond to R7 in arrestin-2. However, we do not believe R7 to be important in phosphorylation recognition, since it is not conserved in arrestin-1 and has never been shown to be a critical phosphosensor in arrestin-2 or arrestin-3 (as far as we are aware). The overall placement of the V2Rpp on arrestin-2 is determined by multiple contacts involving many phosphate groups along the length of the peptide. If the phosphate groups upstream of the key

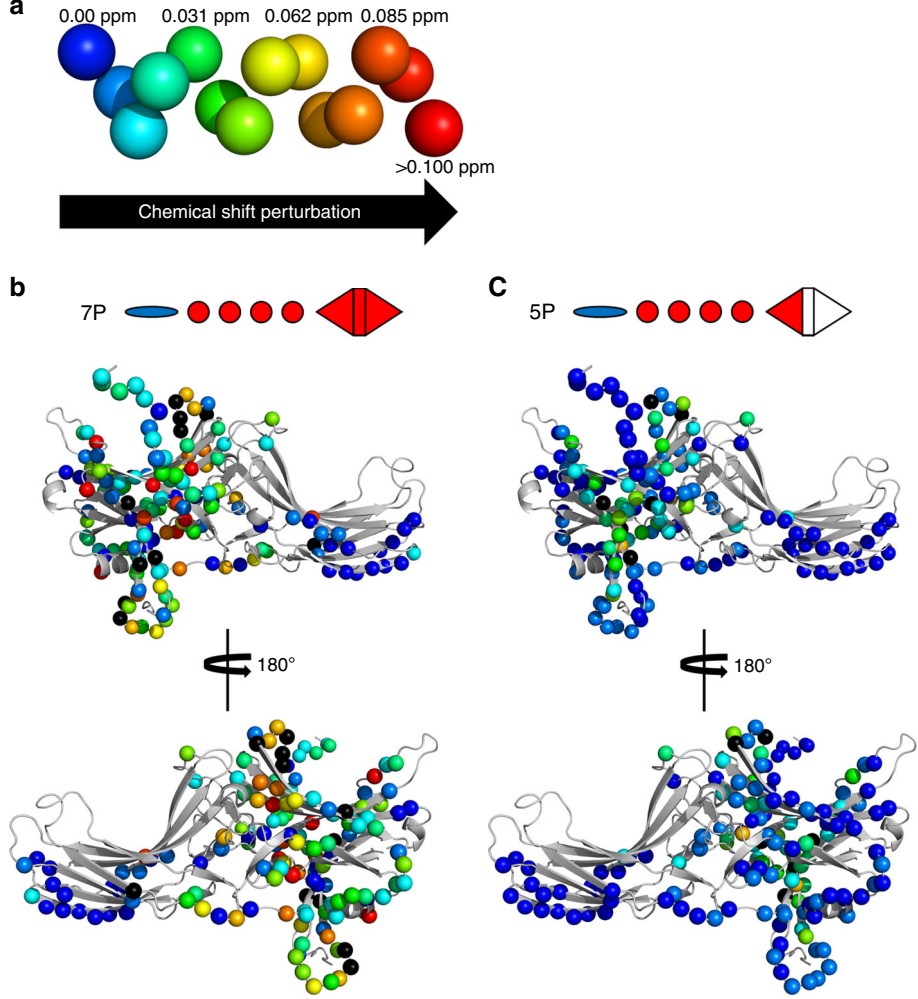

**Fig. 6** Chemical shift perturbations measured by NMR titration series. **a** Color code indicating the magnitude of chemical shift perturbations, which are plotted on structural models of basal arrestin-1 (PDB accession 1CF1, molecule D) in (**b**) and (**c**). The rainbow spectrum represents the range of chemical shift perturbations: nonsignificant (blue), minor (green), moderate (yellow), large (orange), and very large (red). Black marks residues whose signals showed strong line broadening which prevented the final position to be determined, indicating significant changes in mobility of the residue or its surroundings bring them to the μm-ms range. **b** Chemical shift perturbations measured for phosphopeptide 7P. **c** Chemical shift perturbations measured for phosphopeptide 5P. Note that the full-length C-terminus shown in these figures, which is not resolved in the structure, was modeled (see Methods)

sites were absent, the peptide would be free to shift register and sample multiple binding modes. The phosphate at the inhibitory site could then compete with the key sites for binding to their arrestin binding sites and thereby prevent the simultaneous engagement of arrestin near the three-element interaction and the polar core. The spacing between the inhibitory site and either key site is too short to engage both arrestin elements. Thus the modulator phosphorylation sites we identified upstream of the key sites might serve as important anchor points for the phosphorylated receptor C-terminus, to ensure proper placement of the key phosphorylation sites within the arrestin N-domain. This hypothesis would explain why at least two phosphates are necessary for arrestin-1 to bind rhodopsin (the key sites), but three are required for maximal affinity (key sites plus one upstream anchor) [10]. A recent crystal structure of activated arrestin-3 with two molecules of inositol-6-phosphate bound in the N-domain shows that the minimal requirements for arrestin activation, including engagement of the three-element interaction, gate loop and "upstream sites" near the finger loop, could also be fulfilled by polyanions other than phosphopeptide [51].

In the case of arrestin-2 no similar inhibitory effect of site pT342 was observed. The increased propensity of C-tail displacement in this arrestin isoform might allow the active conformation to be stabilized by a few specific interactions with phosphate groups on the peptide. Alternatively, R7 in arrestin-2 could stabilize pT342 (see above) and thereby negate its inhibitory effect. Notably, the inhibitor site is present in the C-termini of many GPCRs (Fig. 8c), which raises the intriguing possibility that arrestin dissociation from the GPCR could be controlled by additional phosphorylation events rather than dephosphorylation by phosphatases. The functional consequences of this possibility for the cellular functions of arrestin have yet to be explored.

Given the high sequence and structure homology of arrestin isoforms, similarities in activation mechanism by different GPCRs are expected. Both the receptor transmembrane helical core and phosphorylated receptor C-terminus contribute to arrestin activation [45,52]. However, our data suggest that the relative contributions of these two receptor components to activation are different for arrestin-2 and arrestin-3. For arrestin-2, the

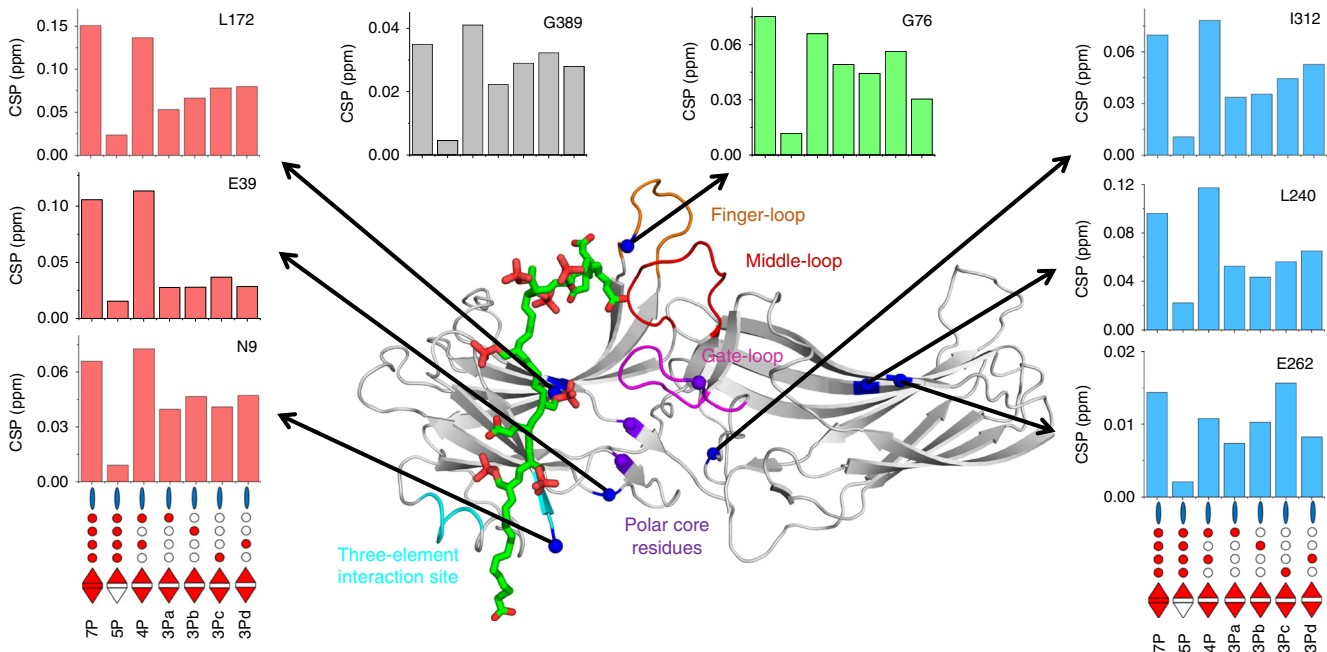

**Fig. 7** Phosphorylation patterns modulate the conformation of arrestin-1. The magnitude of chemical shift perturbations of individual residues caused by different phosphopeptides is illustrated by barplots. The locations of these residues (blue spheres) are indicated on a model of active arrestin-1 in complex with the fully phosphorylated rhodopsin C-terminal peptide (own work; negatively charged and phosphorylated residues are highlighted in red, C-terminus of peptide is at bottom). Note the location of G389 is not shown because the C-tail was not included in the model. The barplots are colored according to residue location in different functional regions of arrestin-1: on or near the polar core (red), C-tail (gray), central crest (green) and C-domain (blue). Additionally, four regions important for arrestin activation are highlighted: the finger loop (orange, G68-L77), the middle-loop (red, Q133-S142), the gate loop (magenta, D296-N305), the three-element interaction (cyan, H10-I13, L107-L111) and the polar core (purple spheres, R175, D30, D303)

C-tail tends to be spontaneously displaced more frequently in the basal state (Fig. 3 and Supplementary Figure 1), which allows arrestin-2 to robustly bind phosphorylated receptor C-termini as long as the key phosphorylation sites are present (Fig. 2c and Table 2). This attribute would explain why arrestin-2 is a relatively poor binder of class A GPCRs[31], whose C-termini generally lack the proper spacing between phosphorylation sites to fulfill the requirement to include both the key sites and negatively charged region[53] (Fig. 8c). Class A GPCRs contain many phosphorylation sites that could potentially fulfill the role of the secondary recruiter sites (pT335 and pS338 in rhodopsin), which could underlie the transient nature of class A interactions[31]. Arrestin-3 binds class A GPCRs better than arrestin-2 [31], and arrestin-3 can even bind GPCRs lacking C-termini[22,54]. These observations indicate that receptor phosphorylation plays a less significant role as compared to the active receptor core in arrestin-3 activation and binding. In contrast to class A GPCRs, the C-termini of class B receptors fulfill the requirements for the key sites and negatively charged motif (Fig. 8c). This combined feature allows these receptors to form tight complexes with arrestin-2 and -3. These differences may be at the root of the distinct signaling profiles of arrestin-2 and -3[16,17,55,56].

The phosphorylation motif we propose here expands upon that which was recently proposed by Zhou et al. based on the crystal structure of the rhodopsin-arrestin-1 complex[25]. Note that species differences do not account for the differences in our proposed motifs, since the bovine (this study) and mouse (Zhou et al.) sequence differ only at site 335 (phosphorylatable threonine in bovine, alanine in mouse). We found that the modulatory site T335 is not critical for arrestin affinity and activation, and it is likely redundant to S334 and T336. The analysis by Zhou et al. was limited to pT336, pS338 and E341, which were resolved on

the C-terminus of the receptor interacting with positively charged pockets on arrestin-1. Despite the insight the structural context affords for this one example of a phosphorylation motif, only a systematic analysis of the effects of different phosphorylation patterns can reveal the regulatory effects of each site on arrestin function. Among the phosphopeptides we analyzed, all but two of the nonbinders (3Pf and 3Ph) contained the barcode motif Zhou et al. proposed on the basis of their structure to be necessary for binding, whereas three of the binders (4P, 3Pa, and 3Pb) did not. The wider phosphorylation motifs we describe here expand not only the size of the coding region but go further in assigning distinct functions to the different regions within the receptor C-terminus. That being said, the current study was (for practical reasons) limited to rhodopsin-derived peptides, and future experiments using peptides derived from other GPCRs will be required to validate the universality of the proposed motifs and to reveal other receptor-specific arrestin binding determinants. For example, in the crystal structure of the rhodopsin-arrestin-1 complex[25], E341, which is positioned between the two key sites, is observed to interact with functionally important charged residues, namely the phosphosensor K15 [48]. This interaction could contribute to the specificity of rhodopsin for arrestin-1 binding[33].

Our proposed phosphorylation motifs are consistent with the multiple and physically extended interactions seen in the crystal structure of arrestin-2 bound to V2Rpp[37], as well as studies showing that arrestin activation and cellular functions are controlled by multiple phosphorylation moieties spread over a large coding region in the GPCR C-terminus[16,17,50]. In summary, the phosphorylation motifs we identify here, which control affinity, activation, inhibition and modulation of global conformation, would support complex regulation mechanisms for the structurally flexible and functionally versatile arrestins.

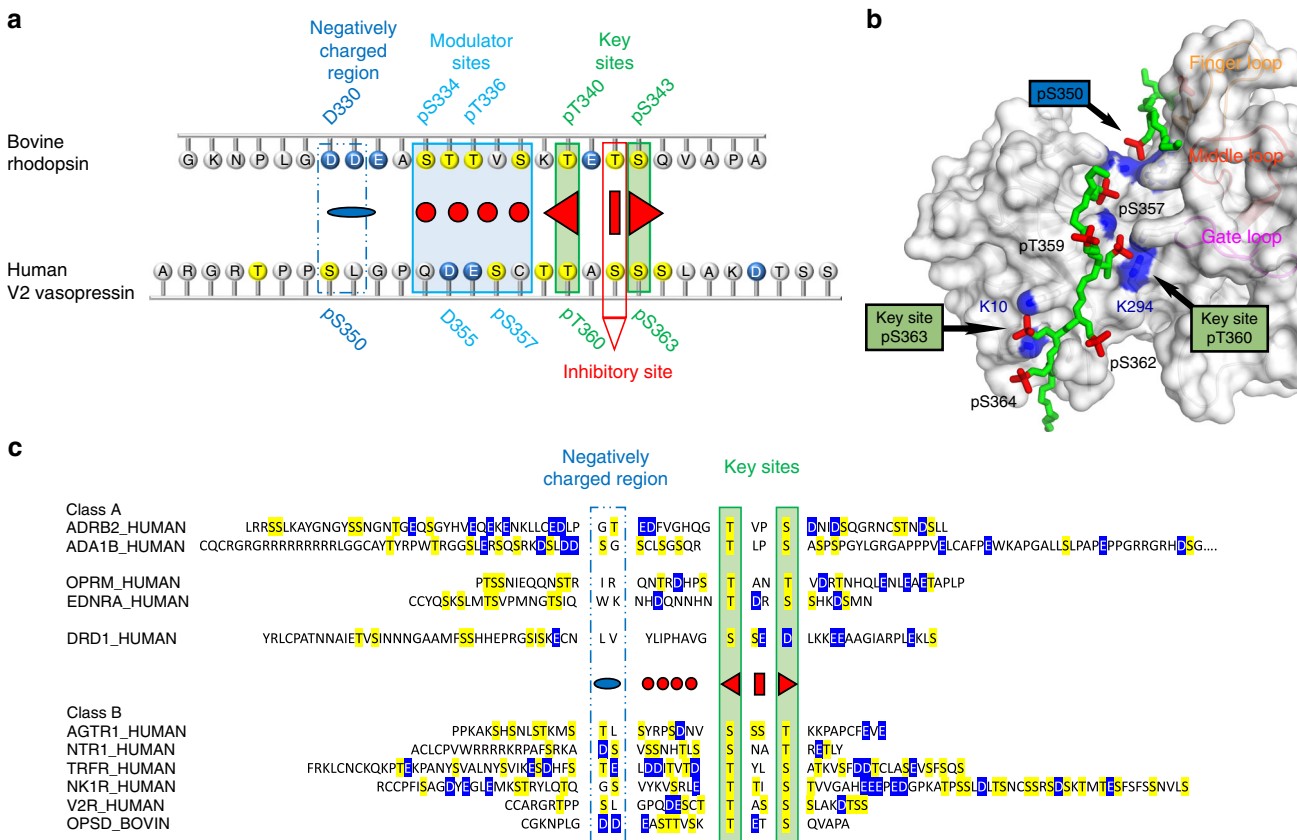

**Fig. 8** Phosphorylation motifs within the larger family of GPCRs. **a** The three functionally distinct phosphorylation motifs described in this study are illustrated on aligned C-termini from two class B GPCRs. Negatively charged amino acids are colored blue and potential phosphorylation sites are colored yellow. **b** Crystal structure of the fully phosphorylated C-terminus of the class B GPCR V2 vasopressin receptor (V2Rpp, green stick model with phosphorylation sites highlighted in red) bound to arrestin-2 (gray surface model) (PDB accession 4JQI)[37]. Important interactions between certain phosphate groups on the peptide (red) and basic residues on arrestin-2 (blue patches) are indicated (see main text for details). **c** Sequence alignments of receptor C-termini from class A and class B GPCRs. All serines and threonines (possible sites of phosphorylation by the GPCR kinases) are highlighted in yellow. Negatively charged residues are highlighted in blue. The sequences are aligned based on the key site motif (serine or threonine residues separated by two amino acids, green box) and the negatively charged motif (two acidic or one phosphorylatable serine or threonine residues 8 residues upstream of the closest activator site, blue box). The alignment indicates that all class B receptors (strong arrestin binders) contain both the key sites and the negatively charged motif. In contrast, class A receptors (weak arrestin binders) generally lack the negatively charged motif (see Supplementary Note 1 for a more detailed description)

## Methods

**Peptide synthesis**. Peptides were synthesized using combinatorial Fmoc Solid Phase Synthesis strategy. The coupling methods applied were specific for each amino acid and were determined based on the phosphorylation pattern as described by Samarasimhareddy et al.[32].

**Constructs**. Synthesized cDNAs of bovine arrestin-1, human arrestin-2 and human arrestin-3 were ordered from Genewiz, amplified by PCR and cloned (Supplementary Information Table 5) into the pET-15b vector (Novagen), using the Gibson Assembly protocol[57]. The constructs contained an N-terminal hexa histidine tag linked by a linker that contained a Tobacco Etch Virus cleavage site (TEV; ENLYFQGGS). PCR was used to introduce the monomerizing arrestin-1 mutations (F85A, F197A)[58]. Note that this mutant was employed for fluorescence anisotropy, trypsin digestion, and NMR experiments. Wild-type arrestin-2 and -3 were used for fluorescence anisotropy and trypsin digestion experiments. For the peptide array experiment, an N-terminal mCherry-arrestin-1 fusion construct was cloned that had a six residue linker (GSGGGS) between the two proteins. The arrestin-1 in this construct had the wild-type sequence. Note that this construct differs slightly from that previously published[59,60] and was optimized for purification. We verified the functionality of this fusion construct using the standard centrifugal pull-down assay using phosphorylated rhodopsin in native ROS membranes[44].

**Arrestin expression**. NiCo (DE3) cells (New England Biolabs) were transformed with plasmid containing the arrestin gene and plated on selective medium. A single colony was used to inoculate 500 ml of lysogenic broth medium, and cells

were grown at 30 °C/160 rpm overnight. Forty milliliter of starting culture was used to inoculate 500 ml of terrific broth media containing 50 μg/ml carbenicillin in a 2 l baffled flask (12×). The culture was incubated in a shaker at 37 °C/160 rpm for 3–4 h until an $OD_{600}$ above 2.0 was reached. The temperature was then reduced to 20 °C and overexpression of arrestins was induced by the addition of 30 μM Isopropyl-β-D-thiogalactopyranosid after 1 h. The cells were harvested on the next day, approx. 20–24 h post induction by centrifugation (3000 × $g$ for 20 min). Pellets were resuspended in 180 ml solubilization buffer (50 mM Bis-Tris propane pH 7.0, 500 mM NaCl and 10% (v/v) glycerol), frozen and stored at −20 °C.

For the expression of $^2$H, $^{15}$N, $^{13}$C-Arrestin-1 the cells were harvested after reaching an $OD_{600}$ of 2.0 by centrifugation (2000 × $g$ for 20 min). The pellets were resuspended immediately afterwards in 2 l M9 media (50 mM $Na_2HPO_4$*2x$H_2O$, 25 mM $KH_2PO_4$, 10 mM NaCl, 2.5 g/l D-Glucose-$^{13}C_6$,1,2,3,4,5,6,6-d$_7$, 1 g/l $^{15}NH_4Cl$, 2 mM $MgSO_4$, 50 μM $FeSO_4$, 2.50 μM $CaCl_2$*2x$H_2O$, 2.07 μM $H_3BO_3$, 150 nM $CoCl_2$*6x$H_2O$, 50 nM $CuCl_2$*2x$H_2O$, 5 μM $ZnCl_2$, 5 μM $Na_2MoO_4$*2x$H_2O$, 400 nM $MnCl_2$*4x$H_2O$, 200 nM Thiamin-hydrochloride, 409 nM D-Biotin, 716 nM choline chloride, 227 nM folic acid, 819 nM niacinamide, 456 nM D-pantothenic acid, 27 nM riboflavin) containing 50 μg/ml carbenicillin. The solution consisting of cells and M9 media was divided into four 2 l baffled flask. They were cultured in the new media for 1 h at 37 °C before they were cooled down to 20 °C. The induction and harvesting was carried out as described for the expression of unlabeled arrestins.

For the expression of the mCherry-arrestin-1 fusion, 1000 μM IPTG was added for induction at 37 °C. The cells were harvested after 2 h post induction by centrifugation (3000 × g for 20 min). Pellets were resuspended in 60 ml solubilization buffer frozen and stored at −20 °C.

**Arrestin purification**. Cell pellets from 6 l of culture were thawed and topped up to a total volume of 240 ml with solubilization buffer. Two tablets of EDTA-free protease inhibitor (Complete-Roche), a small spatula of DNAse (Sigma) and 8 mM 2-mercaptoethanol were freshly added before the cells were lysed by sonication (3 × 5 min, 1 s pulse/0.5 s pause). Ultracentrifugation was performed to remove the insoluble fraction (185,677 × g, 1 h, 4 °C). The supernatant was filtered through 3–4 syringe filters (0.45 μm), and imidazole (30 mM) was added before the lysate was loaded onto a 5 ml Ni-NTA FF crude column (GE Healthcare). The column was washed with ten column volumes of buffer A (20 mM Bis-Tris propane pH 7.0, 500 mM NaCl, 40 mM imidazole, 10% (v/v) glycerol, 8 mM 2-mercaptoethanol). A step elution with IMAC B buffer (same as A but with 500 mM imidazole) was performed. The eluted-fractions were pooled and dialyzed against $SQ_{120}$ (20 mM Bis-Tris propane pH 7.0, 120 mM NaCl, 10% (v/v) glycerol and 14.3 mM 2-mer-captoethanol) overnight in the presence of TEV (10:1 molar ratio of protein to TEV). The dialyzed solution was slowly diluted with $SQ_0$-buffer (0 mM NaCl) to a concentration of 40 mM NaCl. The protein was loaded directly after the dilution onto the SQ column (15 ml resin). The column was washed until the baseline stabilized before switching to the elution buffer ($SQ_{120}$ containing 120 mM NaCl for arrestin-1 or $SQ_{200}$ containing 200 mM NaCl for arrestin-2/-3). Ten-milliliter fractions were collected and protein-containing fractions were pooled. The NaCl concentration was increased to 300 mM before the protein was concentrated with a Vivaspin concentrator (Sartorius, 30 kDa cutoff) to approximately 3 mg/ml. The concentrated protein was flash frozen in 200 μl aliquots and stored at −80 °C.

Purification of mCherry-Arrestin-1 fusion constructs was carried out as described above, except that the volume of the solubilization buffer was reduced to 80 ml and no SQ column purification step was performed.

**ROS preparation**. ROS membranes were prepared from frozen bovine retina obtained from W.L. Lawson Company (USA) or from bovine eyes obtained from a slaughterhouse near Berlin. The American bovine tissue was obtained from animals slaughtered under guidelines set forth by the Humane Slaughter Act (US Public Law 85–765), and all retinal tissue regardless of origin was approved for laboratory use by the State Office for Health and Social Affairs (LAGeSo Berlin). ROS isolation was carried out as previously described[43,61]. Rhodopsin phosphorylation by the native rhodopsin kinase (GRK1) present in ROS was achieved by adding ATP and $MgCl_2$ and illuminating the ROS with light as described previously[43]. Nonphosphorylated ROS membranes were treated identically except that no ATP was added. After illumination 50 mM hydroxylamine was added to the bleached ROS membranes to yield retinal-free opsin. Rhodopsin was regenerated by adding a threefold molar excess of 11-cis-retinal, which was prepared from commercially available all-trans-retinal and purified by HPLC[62]. Regeneration was terminated by the addition of 20 mM o-tert-butyl-hydroxylamine[63]. Rhodopsin concentration was determined by absorbance difference spectrum (extinction coefficient at 500 nm: 40,800/M/cm) after bleaching the rhodopsin in the presence of hydroxylamine.

**Peptide array**. The 384-peptide microarray containing all possible combinatorial phosphorylation patterns from mono- to tri-phosphorylated peptides derived from the rhodopsin C-terminus was ordered from Intavis (Germany). The peptide array was rehydrated by dropwise addition of 96% ethanol to the surface, followed by extensive washing of the surface with deionized water. The blocking of the peptide array was performed with 3% (w/v) bovine serum albumin (BSA) in incubation buffer, which consisted of 20 mM Bis-Tris propane pH 7.0, 250 mM NaCl and 10% (v/v) glycerol. The peptide array was washed twice for 5 min after 1 h of blocking. The peptide array was then incubated with the 5–10 μM mCherry-arrestin-1 at RT for 4 h. The peptide array was rinsed twice with buffer before the bound protein was detected by mCherry fluorescence with automated exposure time setting (Amersham Imager 600). The analysis was performed using the open source software ImageJ and its plugin Protein Array Analyzer, Microsoft Excel and R (www.r-project.org). All peptides were present in duplicate on the array, and two independently synthesized arrays were used; hence, the average signals from four measurements per peptide were used for data analysis. In order to identify the contribution of each phosphorylation site to arrestin-1 binding, two different approaches were used: (1) All peptides were ordered by their fluorescence intensity (binding to arrestin-1) and plotted in a binary heatmap that showed whether a site was phosphorylated or not. (2) A linear regression with an exhaustive, bidirectional, stepwise subset approach (all-model selection) was applied, with the Akaike information criterion to evaluate fits, using the MASS and leaps packages in R. The following fitting function was used, whereby each phosphorylation site was represented by an individual term in the linear regression:

$$y = \theta_1 P334 + \theta_2 P335 + \theta_3 P336 + \theta_4 P338 + \theta_5 P340 + \theta_6 P342 + \theta_7 P343. \quad (1)$$

As some peptides can show low fluorescence signals, the same approach was repeated using varying thresholds to filter low signal peptides and check the robustness of the most important positions.

**Fluorescent labeling of N-terminal peptide cysteine**. 0.1–0.2 mg of lyophilized peptide was dissolved in 50 μl of 50 mM phosphate-buffered saline (PBS) buffer at pH 7.0, and then two molar equivalents of fluorescein-5-maleimide dye stock dissolved in dimethylsulfoxid was added. The reaction was carried out at RT in the dark. After 1 h the reaction was topped up with 1 ml of ice-cold 100% acetone and stored until the next day at −20 °C. The precipitated peptide was centrifuged (13,000 × g) for 10 min. The labeled and precipitated peptide was sedimented, and the supernatant was removed. The peptide was resuspended again in ice-cold acetone, stored for 1 h at −20 °C and centrifuged (13,000 × g) for 10 min. This procedure was repeated until the supernatant did not contain any color when it was diluted with 50 mM PBS buffer, pH 7.0. The precipitated peptide was dried at RT in the dark. It was dissolved in 50 μl of 20 mM Bis-Tris propane pH 7.0 and 150 mM NaCl and its concentration was determined by absorbance. A molar extinction coefficient of 70,000/M/cm at pH 7.0 was assumed for calculating the concentration of the labeled peptide stock.

**Fluorescence anisotropy**. An 11-step serial dilution series of arrestin in 20 mM Bis-Tris propane pH 7.0, 150 mM NaCl, 8 mM 2-mercaptoethanol, 2% (w/w) BSA, 0.02% (v/v) Tween-20 was prepared and transferred to a black high-bottom 384-well plate (Greiner). As control for free peptide, a position with only buffer was included. The peptide was added at a final concentration of 20 nM to each well. The plate was closed by a clear cover and vortexed for 15 s. Afterwards it was centrifuged (800 × g) for 2 min and stored at RT for 20 min. The reading was performed by default settings of PHERAstar plate-reader (BMG Labtech) and excitation-/emission-wavelengths of 485 nm (12 nm bandpass)/520 nm (30 nm bandpass) were used. The analysis was performed using Origin2016 (Microcal Inc) software by fitting the data to a one-to-one binding model:

$$r = R_0 + \triangle R * \left( \frac{P}{K_d + P} \right), \quad (2)$$

where r stands for anisotropy, $R_0$ stands for anisotropy of the ligand, $\Delta R$ stands for anisotropy of the complex minus anisotropy of the ligand, P stands for the concentration of protein, and $K_d$ stands for the dissociation constant of the ligand.

**Limited trypsin digestion of arrestin**. Arrestin (60 μg) was mixed at a 1 to 1.2 molar ratio with peptide and preincubated at RT for 5 min before 1 μl of 0.15 mg/ml trypsin stock was added. The enzymatic digestion was carried out at 35 °C (20 mM Bis-Tris propane pH 7.0, 150 mM NaCl, 10 mM 2-Mercaptoethanol and 2 mM EDTA). At certain time points (5, 15, 30, 60, 90 min), 5 μl aliquots were quenched with heat incubation in the presence of 20 μl SDS loading buffer (95 °C, 5 min) and then subjected to SDS PAGE. Bands were visualized with InstantBlue (Expedeon).

**Site-directed fluorescence and pull-down experiments**. Recombinant bovine arrestin-1 cloned into the pET15b vector was expressed and purified as described[43]. The basic construct was a modified wild-type sequence that contained no native cysteines (C63A, C128S and C143A) nor tryptophans (W194F). Single cysteine residues were introduced by PCR at sites 72 or 197 as previously described[43]. The I72C mutant was labeled with monobromobimane (Thermo Fisher) and is referred to as I72B. The F197C mutant was labeled with N,N′-Dimethyl-N-(Iodoacetyl)-N′-(7-Nitrobenz-2-Oxa-1,3-Diazol-4-yl)Ethylenediamine (IANBD amide, Thermo Fisher) and is referred to as F197NBD. Labeling was carried out as previously described[43,61]. Both labeled mutants were assessed to be functionally similar to wild-type arrestin-1 using the Extra Meta II assay, which determines the ability of arrestin-1 to bind and stabilize the active Meta II form of rhodopsin[64].

Under dim-red light, samples were prepared that contained nonphosphorylated rhodopsin (10 μM) in ROS membranes, 1 μM arrestin-1 I72B or F197NBD, ±100 μM peptide in 50 mM HEPES, 130 mM NaCl, pH 7 (150 or 200 μl sample volume). One hundred microliters of the sample was used for fluorescence measurements, and the remainder was used for the centrifugal pull-down assay (50 μl per assay). Steady-state fluorescence measurements were performed using a SPEX Fluorolog (1680) instrument in front-face mode. Bimane fluorescence was recorded using an excitation wavelength of 400 nm and emission 424–600 nm, and NBD fluorescence was recorded using an excitation wavelength of 500 nm and emission 524–660 (2 nm step size, 0.5 s integration per point). Excitation slits were narrowed to 0.1–0.2 nm to minimize light-activation of rhodopsin, and emission slits were set at 4 nm. Pull-down analysis was performed by centrifuging samples at 16,000 × g for 10 min at 20 °C. Selected samples were light activated (>495 nm) for 15 s prior to centrifugation (appr. 60 s delay time). Following centrifugation, the supernatants were removed, and the pellets were solubilized in loading buffer containing 2% SDS and subjected to SDS PAGE. Bands were visualized using Coomassie Brilliant Blue.

**Statistical analysis**. The statistical analysis of the affinity measurements, the site-directed fluorescence spectroscopy experiments and pull-down experiments were performed in Prism6 using ANOVA analysis. The significance between two values is given either by "ns" = no statistical significance, *$P \leq 0.05$, **$P \leq 0.01$, ***$P \leq 0.001$ and ****$P \leq 0.0001$ or '−' if no calculation was possible due to no measurable $K_d$ value in case of fluorescence anisotropy phosphopeptide binding assays.

**NMR sample preparation.** $^2$H,$^{15}$N,$^{13}$C-labeled arrestin-1 aliquots were thawed quickly under running water, transferred to 30 kDa-cutoff Vivaspin (0.5 ml, Sartorius) centrifugal spin concentrators, and concentrated to approximately 600 μM. The concentrator was immediately transferred into a beaker filled with 400 ml of NMR buffer 1 or 2 (see below) carried by a floatation device. The dialysis was performed at 4 °C overnight. On the next day the concentration was determined by UV absorbance at 280 nm (molar extinction coefficient 26,360/mol/cm). Lyophilized peptides were dissolved in NMR buffer 2 and dialyzed with a self-made dialyzing device consisting of the top of a 1.5 ml Eppendorf tube and a dialysis membrane (100–500 Da cutoff, Biotech CE Tubing, Spectrum LABS). The dialysis was performed at 4 °C overnight, and the peptide concentration was determined by its absorbance at 205 nm (molar extinction coefficient: 54,310/mol/cm). The NMR sample was mixed and 5% $D_2O$ was added. Afterwards it was transferred to a 3 mm Shigemi tube using a plastic syringe connected to a piece of Teflon high performance chromatography tubing using a luer lock (Pharmacia) and sealed with a piece of Parafilm M. NMR buffer 1: 25 mM Bis-Tris propane pH 7.0, 150 mM NaCl, 10 mM DTT and 0.02% $NaN_3$; NMR buffer 2: 350 mM MOPS (adjusted with Bis-Tris propane) pH 7.0, 5 mM $NaH_2PO_4$, 14.3 mM 2-mercaptoethanol and 0.02% $NaN_3$. Buffer 2 was necessary for titrations experiments to avoid systematic changes in pH due to addition of phosphopeptides.

**NMR measurements and data analysis.** NMR measurements were performed on Bruker Avance III-HD 600 or 900 MHz spectrometer using a CTCI cryo-cooled probehead. At the beginning of all experimental series the temperature was calibrated to 298 K using a sample of 99.8% methanol-$d_4$ (Bruker) in a 5 mm NMR tube. Pulse lengths were calibrated for $^1$H manually and for $^{13}$C/$^{15}$N using an in-house automated procedure. pH was monitored using the position of the phosphate signal in NMR buffer 2 as an internal reference and varied by no more than 0.03 pH units during peptide titration experiments. [$^{15}$N,$^1$H]TROSY experiments were recorded with internal ETH pulse sequences, whereas 3D triple resonance TROSY experiments were recorded using the manufacturer's pulse sequences. Spectra were processed in TopSpin 3.2 (Bruker) and calibrated using external 4,4-dimethyl-4-silapentane-1-sulfonic acid in the NMR buffer. All further data analysis was performed in CARA[65] (www.nmr.ch), Excel and Origin.

**NMR resonance assignments.** The previously published assignments of arrestin-1[66] were used as basis for the assignments in this work. Minor differences in peak positions were observed due to the use of a different buffer composition, but for most resonances only slight shifts in peak positions were observed. Assignments were confirmed using triple resonance spectra measured with samples with protein concentration of 350–600 μM. The backbone amide resonance assignment had a completeness of approximately 40%.

**NMR titrations.** The concentration of arrestin-1 was held constant at 100 μM and the peptide was added at stoichiometric arrestin-1:peptide ratios of 1:0 (protein alone), 1:1, 1:4 and 1:10. Each titration series consisted of four [$^{15}$N,$^1$H]TROSY experiments, where the sample from the previous step of the titration series was recycled and topped up with a peptide stock (5 mM) as well as arrestin-1 to reach the next aimed stoichiometric ratio at a constant arrestin concentration. As a control, a [$^{15}$N,$^1$H]TROSY spectrum containing arrestin-1 and nonphosphorylated C-terminal peptide (0P) at a 1:10 molar ratio was measured, which showed some minor changes in peak intensity for a small set of signals when compared to the corresponding spectrum obtained with protein alone. These changes in intensity may be caused by weak interactions mediated by four acidic residues (two aspartates and two glutamates) in the peptide sequence. The position of signals was followed from their starting point at 0:1 peptide:arrestin-1 ratio up to the endpoint at 10:1 ratio. TROSYs were measured for 18–48 h each on a 600 MHz spectrometer.

**NMR data analysis.** The chemical shift perturbation of each $^1$H-$^{15}$N signal between the [$^{15}$N,$^1$H]TROSY spectra obtained with a given peptide:protein ratio and the spectrum of the reference protein alone was calculated using the formula:

$$\Delta\delta = [(0.2\times(\delta_N^{peptide+protein} - \delta_N^{protein}))^2 + (\delta_H^{peptide+protein} - \delta_H^{protein})^2]^{1/2}, \quad (3)$$

where $\delta_N^{protein}$ is the chemical shift of the amide $^{15}$N signal for the protein alone, $\delta_H^{protein}$ is the corresponding chemical shift for the $^1$H signal, and the corresponding values for the protein in the presence of peptide are indicated by the same symbols with superscript peptide + protein.

**Modeling of full-length arrestin-1.** The program MODELLER9.14 was used to create a full-length representation of arrestin-1 (C-terminal sequence taken from Uniport:P08168). A total of 100 models were created and subjected to 300 iterations each. One of the top scoring models using a discrete optimized protein energy potential was selected by eye and equilibrated using the Implicit solvent modeler GBSW with the CHARM-Gui and NAMD2.10.

**Reporting summary.** Further information on experimental design is available in the Nature Research Reporting Summary linked to this article.

### Data availability
Data supporting the findings of this manuscript are available from the corresponding authors upon reasonable request. A reporting summary for this Article is available as a Supplementary Information file. The source data underlying Figs. 2c, d, 3, 4c, d and Supplementary Figures 1 are provided as a Source Data file.

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

## Acknowledgements

We thank Miriam Zimmermann, Florian Wilhelm, David Sykes, A.J. Venkatakrishnan and Madan Babu for the critical reading of the manuscript, and Timothy Sharpe and Ben Schuler for help and advice with peptide binding experiments. This work was supported by the Swiss National Science Foundation grants 135754 and 159748 to D.B.V., 310030_153145 and 310030B_173335 to G.F.X.S., SNF Sinergia CRSII3_141898 to D.B. V., F.H.T.A. and G.F.X.S., the Israeli Science Foundation 939/14 to A.F., the Deutsche Forschungsgemeinschaft SO1037/1-3 to M.E.S., the Berlin Institute of Health (Delbrück Fellowship BIH_PRO_314 to M.E.S.), the ETH Zürich postdoctoral fellowship to T.F., the European Union Horizon 2020 Programme Grant 664786 to M.H., and the Hebrew University (Lady Davies Postdoctoral Fellowship to M.S.). Many of the authors participated in the European COST Action CM1207 (GLISTEN).

## Author contributions

F.F.D., H.-T., G.F.X.S. and D.B.V. conceived the project, with input from J.S. D.M., M.F., B.B., and Z.V. prepared protein reagents. M.S., M.H., and A.F. developed the synthesis strategy and prepared phosphopeptides. D.M. performed peptide array and trypsin digest experiments. D.M. and T.F. analyzed the peptide array data. D.M., M.F., and F.Z. performed peptide binding experiments. M.E.S. and D.M. performed parallel site-specific fluorescence and pull-down experiments. D.M. and F.F.D. performed the NMR experiments and analyzed the data. E.M. and X.D. built homology models of arrestin-1 bound to phosphopeptide/phosphorylated receptor. F.F.D. contributed NMR expertise, M.E.S. contributed arrestin activation expertise, M.H. and A.F. peptide synthesis expertise. D.M., F.F.D., M.E.S., and D.B.V. wrote the manuscript with input from co-authors. D.B.V. supervised the project.

## Additional information

**Competing interests:** The authors declare no competing interests.

