## [Peer Review File · Nature Communications]

FIRST ROUND

Reviewers' comments:

Reviewer #1 (Remarks to the Author):

This is a study of the role of rhodopsin phosphorylation sites in arrestin-1 binding. The authors used an array of differentially phosphorylated peptides derived from the C-terminus of bovine rhodopsin to determine their binding to arrestin-1, as well as arrestin-2 and arrestin-3, their ability to induce conformational changes in arrestin-1 (which is usually regarded as arrestin activation), and to facilitate arrestin-1 binding to unphosphorylated light-activated rhodopsin. The study is remarkably comprehensive, the data are pretty strong and quantitative, but some discrepancies with the previously published data should be addressed in discussion.

1. Human and mouse rhodopsin has 6 phosphorylation sites, whereas bovine has seven. The authors used peptides of the bovine rhodopsin, whereas Zhuo et al (ref 22) used human rhodopsin and mouse arrestin-1 in their structural work. To what extent the discrepancy between the phospho-code identified here (Fig. 8) and in ref 22 can be explained by species differences? Previous findings that human (Invest Ophthalmol Vis Sci. 2018 Jan 1;59(1):13-20) and mouse (Biochemistry. 2011 Mar 29;50(12):2235-42) arrestin-1 binds bovine phosphorylated rhodopsin essentially as well as bovine arrestin-1 suggest that the key players must be common across species. Explain.

The bovine and human rhodopsin C-terminus sequences differ only at position 335 (Thr in bovine and Ala in human). As our study demonstrates, T335 in bovine rhodopsin is not as critical as sites T340 and S343 for recruiting arrestin with high affinity, and plays a secondary role: to enhance arrestin affinity and activation in the absence of phosphorylation at one of the key sites. Hence it is not surprising that human and mouse arrestin-1 bind bovine rhodopsin robustly (assuming the key sites are phosphorylated in addition to one upstream site).

The discrepancy between the phosphorylation code proposed by us and the one proposed by Zhou et al. is not due to species differences. The presence of a threonine at site 335 instead of alanine would not have dramatically altered the interpretation of Zhou et al. They proposed a code of *PxPxxP/E/D* based on the observable sites in their crystal structure (pT336, pS338 and E341). However, we found that some phosphopeptides that do not contain this code robustly bound arrestin, and some peptides containing this code did not bind arrestin (see p. 15 of the manuscript). The code proposed by Zhou et al. fails to predict the binding efficacy of these phosphopeptides since it does not describe the relative contribution of each site to arrestin affinity and activation. Our proposed phosphorylation code explains the experimental data (both from this study and others) by showing that the number and spacing of phosphorylation sites is not as important as which sites are phosphorylated.

We have added the following sentence to the discussion on p14/15 to address the reviewers concern regarding C-termini derived from different species:

"Note that species differences do not account for the differences in our proposed motifs, since the bovine (this study) and mouse (Zhou et al.) sequence differ only at site 335 (phosphorylatable threonine in bovine, alanine in mouse). We found that the

modulatory site T335 is not critical for arrestin affinity and activation, and it is likely redundant to S334 and T336.”

2. The results here (arguably more convincing) do not agree with the previous finding that threonines in the mouse rhodopsin are more important for arrestin-1 binding than serines in vivo (ref 26). The authors should discuss this issue.

There are several reasons why our study and that by Azevedo et al. are not directly comparable. Firstly, ours was an in vitro experimental study using purified components. The study by Azevedo et al. was performed in living mice. Living rod cells represent a much more complicated system than ours, and many factors could have influenced the results. In our experiments, the phosphorylation pattern on each peptide was strictly controlled during synthesis. In the study by Azevedo, potential sites of phosphorylation were removed by genetic manipulation, and phosphorylation of the remaining sites depended on the GRK1 and other cellular machinery present in the rod cells. Notably, threonine-only rhodopsin was not effectively phosphorylated in vivo (>75% 0 or 1 phosphates; >50% no phosphorylation), whereas a majority of the serine-only rhodopsin contained 2 to 3 phosphates (see Fig. 3B of Azevedo et al.). The authors state that the lack of serines inhibits the phosphorylation of threonine-only rhodopsin, since serines are “more rapidly and efficiently phosphorylated than threonine sites”. This lack of phosphorylation casts doubt on the role of threonine phosphorylation in controlling single photon responses (SPR) from mouse rods. Moreover, Azevedo et al. report that threonine-only rod responses are not sensitive to GRK1/arrestin concentration, whereas serine-only rod responses are sensitive to GRK1/arrestin concentration. These results suggest that some other mechanism independent of rhodopsin phosphorylation and arrestin binding might control the SPR in threonine-only rod cells.

In our study, we focus on the molecular mechanism of arrestin interaction with phosphorylated peptides. Our results clearly show different functions for the different sites. How these findings translate into arrestin interactions with different GPCRs in different cell types will be the subject of many future studies.

We have added the following paragraph to the discussion (p.12):

“Two previous in situ studies have suggested that phosphorylated threonines may be more important for arrestin recruitment to rhodopsin than phosphorylated serines (Mendez, Burns et al. 2000, Azevedo, Doan et al. 2015). However, these studies cannot be directly compared to ours, since the exact levels of phosphorylation could not be controlled. Potential sites of phosphorylation were removed genetically, and phosphorylation of the remaining sites depended on the native rhodopsin kinase (GRK1) in the rod cells. Notably, rhodopsin containing only threonines was not effectively phosphorylated (>50% not phosphorylated at all), and its deactivation in the rod cell was apparently independent of GRK1 and arrestin-1 (Azevedo, Doan et al. 2015). These results suggest that other, arrestin-1-independent mechanisms might exist in the rod cell to deactivate rhodopsin.”

3. In vivo study of the sites on mouse rhodopsin necessary for arrestin-1 binding (Neuron. 2000 Oct;28(1):153-64) does not quite agree with the data presented here. The authors should discuss this issue.

In the study by Mendez et al., very prolonged SPR were observed for rhodopsin mutants lacking all phosphorylation sites (CSM). 'Adding back' S338 or S334+S338 did not restore the SPR to wild-type. On the other hand, the S338A or S343A mutant rhodopsins, or threonine-only rhodopsin (STM, containing S334A, S338A, and S343A), gave SPR more like wild-type. Based on these results, the authors conclude that three phosphorylation sites are optimal for fast and timely desensitization of rhodopsin by arrestin.

For the most part, our results and proposed phosphorylation code explain the results reported by Mendez et al. For example, the re-introduction of S338 and S334 into the CSM construct did not restore the SPR, consistent with our finding that these sites are not key for arrestin affinity and activation. On the other hand, we predict that if the authors had re-introduced T340 and S343 into their CSM construct, they would have observed a nearly normal SPR and thus identified the key sites.

Furthermore, removal of site S338 did not affect the SPR, as we found this site to be a 'secondary recruiter'. However, when one of the key sites was removed (S343A), a normal SPR was observed. Our study shows this is expected (indeed the S343A mutant is similar to our peptide 5P), as long as the secondary recruiter sites were phosphorylated in vivo. Notably, mouse rhodopsin lacks one of the secondary recruiter sites (T335 is alanine). We hypothesize that in the absence of T335, the neighbouring sites S334 and T336 might fulfill the role of secondary recruiter site. The one result we cannot explain is the near-normal SPR of mouse rods containing threonine-only rhodopsin. As we explain in the point above, this discrepancy might be due to redundant, unknown desensitization mechanisms in the rod cell. In the revised version of the manuscript, we have addressed the discrepancy with the Mendez et al. study in the same paragraph in which we address the Azevedo et al. study (comment #2 above).

4. Arrestin-3 was recently crystallized in active (receptor bound-like) conformation in complex with IP6 (Nat Commun. 2017 Nov 10;8(1):1427). Does the engagement of the positive arrestin-3 charges by the phosphates in IP6 and the engagement of arrestin-2 positive charges by the phosphates in the angiotensin receptor C-terminus (ref 39) correspond to the phospho-code described here (Fig. 8)? This should be discussed.

We already describe how the crystal structure of arrestin-2 bound to fully-phosphorylated peptide analogue of the C-terminus of the V2 vasopressin receptor (Shukla et al. Nat Comm 2013, Ref. 39) fits to our proposed phosphorylation code (Figure 9). Indeed, we use this crystal structure to describe how the different classes of phosphorylation sites identified in our study interact with and manipulate the structure of arrestin (beginning of Discussion, p. 9 of manuscript).

We did not cite the study by Chen et al. on IP6-bound arrestin-3, since the V2Rpp-Arr2 structure is a more suitable model for interpreting our data using phosphopeptides. That being said, the IP6-Arr3 structure shows engagement of two molecules of IP in the arrestin-3 N-domain. One IP6 molecule is bound near the

'three-element interaction' (e.g. K108, K11), similar to the site bound by key site pS363 in V2Rpp (pS343 in rhodopsin). The second molecule of IP6 interacts with positively charged residues from the polar core and the 160-loop, which is reminiscent of where the key site pT360 and the upstream site pS350 bind on the V2Rpp-Arr2 structure. This second interaction also releases the finger loop from its restricted basal conformation, thereby allowing it to interact with other finger loops found within the Arr3 trimer. Notably, the combined engagement of the three-element interaction, polar core and finger loop by the two molecules of IP6 help to stabilize an active conformation of arrestin in a similar way as the minimally activating combination of phosphorylated sites predicted by our phosphorylation code (two key sites plus one upstream site).

We have added text to the Discussion on p. 13 of the manuscript addressing this point:

"A recent crystal structure of activated arrestin-3 with two molecules of inositol-6-phosphate bound in the N-domain shows that the minimal requirements for arrestin activation, including engagement of the three-element interaction, gate loop and 'upstream sites' near the finger loop, could also be fulfilled by polyanions other than phosphopeptide(29127291)."

5. How were the NMR peaks in arrestin-1 assigned? If the authors used existing assignment (e.g., Proc Natl Acad Sci U S A. 2013 Jan 15;110(3):942-7), the source should be referenced in the main paper.

Resonance assignments reported by Zhuang et al. Biochemistry 2010 were used in our study, and we state this in the Methods. In order to avoid the confusion, we have added this reference and clarifying text to the Results section on p. 9.

6. On the full NMR spectrum (Fig. S2) indicate the peaks enlarged in Fig. 5.

We highlighted the residue peaks as suggested in Supplementary Fig. S3.

7. Reference the source of info that F85A, F197A mutations monomerize arrestin-1.

We cite Hanson et al. Structure 2008 in the Methods when referring to the monomerizing mutations.

8. The authors imply that arrestin-2 is more flexible than arrestins-3, which does not agree with the structural data (J Mol Biol. 2011 Feb 25;406(3):467-78) or molecular dynamics simulations (ACS Chem Neurosci. 2016 Sep 21;7(9):1212-24). Please explain.

In our work we use the term 'flexibility' to describe the relative susceptibilities of the different arrestin isoforms to limited trypsin digest, as they reflect spontaneous activation of arrestin. Trypsin primarily cleaves at the C-terminus of arrestin-1 within the first minutes (Palczewski et al. 1991 JBC v. 255, 15334), and displacement of the C-tail by polyanion binding makes cleavage of the C-tail more efficient and complete (Palczewski et al. 1991 JBC v. 266, 18649). In our hands, arrestin-2 was efficiently digested - even in the absence of phosphopeptide - indicating that the C-tail of arrestin-2 is less tightly anchored to the N-domain and therefore more accessible to trypsin than in arrestin-1 and arrestin-2. This increased flexibility of the arrestin-2 C-tail allows arrestin-2 to bind

phosphopeptides with higher affinity than arrestin-1 and arrestin-3, since C-tail displacement exposes the putative phosphopeptide binding site (Shukla et al: Nature 2013).

The higher structural flexibility in the arrestin-3 beta-sandwich (reported in the two studies cited here by the reviewer) explains why the binding of arrestin-3 to active GPCRs is less dependent on receptor phosphorylation. It was proposed that the increased flexibility around the central crest lowers the energy barrier for arrestin-3 coupling to the active receptor core (see Latorraca et al. Nature 2018).

We have modified the language in the revised manuscript to more clearly indicate the type of flexibility we mean. Specifically:

(p. 6/7) "In the case of arrestin-2, the application of limited amounts of trypsin resulted in relatively fast and efficient digestion, and the pattern of digested protein bands was not significantly changed by the presence of phosphopeptide (Figure 3a). This difference in protease sensitivity likely reflects an increased mobility and spontaneous displacement of the C-tail in arrestin-2. This increased C-tail flexibility allows arrestin-2 to bind phosphopeptides with higher affinity than arrestin-1 and arrestin-3 (Figure 2c, Table 2), since C-tail displacement exposes the putative phosphopeptide binding site(Shukla et al: Nature 2013)."

(p. 14) "For arrestin-2, the C-tail tends to be spontaneously displaced more frequently in the basal state (Figure 3 and Supplementary Figure S1), which allows arrestin-2 to robustly bind phosphorylated receptor C-termini as long as the key phosphorylation sites are present (Figure 2c and Table 2)."

Reviewer #2 (Remarks to the Author):

The manuscript is clearly written, and the experimental work is well done and convincing. Based on a library of synthetic phosphopeptides mimicking different phosphorylation states of the C-terminus of rhodopsin, the arrestin affinity, activation and conformational modulation were measured by several methods. The results indicate that the particular phosphorylation sites can be grouped into three different functional classes: 'key sites,' an 'inhibitory site' and 'modulator sites.' The functional motifs proposed here provide a new insight into the mechanism of arrestin-mediated GPCR functions that could not be described very well by the crystallography.

Introduction and discussion are good, and the authors highlighted the aims and significance. Most of the conclusions are supported by the data presented. There are several things that the authors should consider in the future.

1. There are no error bars in Fig. 2d. How many times have these experiments been done?

There were error bars in the original version of the manuscript, but they were so small they were hidden behind the data points. In the revised version, we have plotted all data from the three independent experiments, so the small measurement error can be easily appreciated by the reader. The legend for Figure 2 has also been modified to clearly indicate this.

2. In supplements Fig S1, Please ensure that all gels are accompanied by the locations of molecular weight/size markers. The bottom figure in Supp Fig. 1b was cropped shorter than others, and please show the full gel.

The revised Supplementary Figure S2 (previously Figure S1) now shows uncropped gels, and the molecular weight marker is now appropriately labelled. Please note, molecular weight marker was run on only one gel in each series due to the limited number of lanes, and we state this now in the legend.

3. The figure 8 and figure 9 may be merged to show the conclusion.

We have merged the figures as suggested.

4. Site-specific fluorescently labelled mutants of arrestin-1 to evaluate the potential of the different phosphopeptides is a good idea. There is one latest paper published last month (<https://www.nature.com/articles/s41589-018-0115-3>), and I suggest it will be cited in this part.

We thank the reviewer for suggesting this reference, which we missed since it was published about the same time we submitted our manuscript. However, we do not think it appropriate to cite this reference here, since the use of fluorescently labelled arrestin mutants to monitor arrestin activation and receptor binding was established long before this recent study. The arrestin I72B mutant was first reported in Sommer et al. 2005 JBC v. 280: 6861-71, and the fluorescently labelled F197C arrestin mutant was first reported in Lally et al. 2017 Nat Comms v. 8: 14258. In contrast, in the study by Yang et al. SRC was labelled with TA-bimane in order to monitor conformational changes in SRC due to beta-arrestin-1 binding. The study by Yang et al. 2018 is still highly relevant to our study, as another demonstration of how different receptor phosphorylation patterns modulate the structure of arrestin. Therefore we now cite this study in the introduction to our paper.

5. The manuscript was well-written. There are several typo errors.

In page3, "a variety cellular outcomes" should be "a variety cellular of outcomes."

Corrected to "a variety of cellular outcomes".

In page4, "provide a framework" should be "provides a framework."

This comment is no longer relevant, since the language at the end of the Introduction has been changed.

In page8, "The magnitude of the chemical shift changes of G389 were" should be "was."

Corrected

In my opinion, it should be accepted after minor revision.

Reviewer #3 (Remarks to the Author):

General Comment: this is an interesting paper dissecting the phosphorylation motifs on a model GPCR, rhodopsin, and using a variety of biophysical techniques to assess how different phosphopeptides affect the conformation of arrestin 1, and the beta arrestins (aka arrestin 2 and 3). The studies are carefully and comprehensively performed but the presentation is confusing in a number of places and also makes claims that are not supported by the data presented. The paper will be of interest to the GPCR community and should be publishable after significant revision.

A major issue in the paper is that the authors' conclusions go far beyond what they have actually shown and need to be dramatically toned down. The concept of a "barcode" of phosphorylation sites on the C termini of GPCRs has gained acceptance over the past 15 years or so. But this paper does virtually nothing to elucidate it. The idea of a barcode directly implies the identification of specific phosphorylation motifs which determine specific functions of arrestins (see definition below under referencing). This growing list of arrestin functionalities includes desensitization, endocytosis and all manner of signaling, both G-protein dependent and independent. But there is no functional work done here and thus one cannot ascribe to any specific phosphorylation pattern corresponding functionalities of arrestin. In this context, I would like the reviewers to defend their use of the rhodopsin-arrestin system as their model for these studies. Other than desensitization of rhodopsin-transducin signaling, there has been virtually no work in the visual system on the expanding list of arrestin functions such as endocytosis and numerous signaling pathways. Hence one might have thought that a much more appropriate system would have been some other GPCR and arrestin 2 and 3 coupled to an attempt to correlate phosphorylation patterns with specific cellular functions. Thus, the conclusion of the authors that "the GPCR phosphorylation motifs we identify here elucidate the barcode that controls the cellular functions of arrestins" is quite without merit and all such statements in the paper need to be toned down. The title of the paper is in fact a valid statement of what is shown, which is much more limited than elucidating any barcode or why some GPCRs interact transiently with arrestins and form stable long-lived complexes. Any attempt to understand this would have to involve consideration of the role of the central helical core of the receptors as well as the phosphorylation pattern on the C terminus.

We thank the reviewer for pointing out our potentially misleading language. In the original version of the manuscript, we used the word 'functional' in the limited sense of how the different phosphorylation sites and motifs affect the arrestin structure and its ability to interact with the phosphopeptide and receptor core. Indeed this was all we could claim based on our data. As requested by the reviewer, we have toned down our language and now make no claim that our data elucidates the bar code. As much as possible, we have tried to clearly state exactly what our data means and its limitations as a purely in vitro biophysical study.

In order to understand how GPCR phosphorylation patterns are translated into a distinct cellular responses, we must first understand how each phosphorylation site affects arrestin on the molecular level. This is the missing link between phosphorylation-pattern-determined arrestin conformations and the cellular functions of arrestin (or 'bar code', as first proposed by Kim et al. PNAS v.102(5) 2005). Only a handful of studies so far have attempted to link phosphorylation-dependent arrestin conformations to cellular functions (e.g. Nobles et al. *Sci Signal*

v.4(185) 2011, Yang et al. *Nat Comms* v.6 2015, Yang et al. *Nat Chem Biol* v. 14(9) 2018). In contrast to these studies, our reductive approach allowed each phosphorylation site to be assigned a specific molecular function (i.e. affinity, activation, conformational modulation). With this knowledge in hand, other scientists in the field will be able to better predict how different GPCR phosphorylation patterns will affect arrestin recruitment, affinity, how long arrestin stays bound to the receptor, the activation state of arrestin, etc. We wish to stress - the conformation of arrestin is not the only factor determining how it interacts with the GPCR (i.e. binding modes) or scaffolds other cellular proteins. Timing - as in how long it stays bound at a receptor or how long it remains in the active conformation - are also critically important. However, all available literature has focused on measuring global conformation and not physical constants like affinity. Fundamentally this was the gap in knowledge our study sought to fill. We chose rhodopsin as a model GPCR, since the most literature is available for this receptor regarding the role of the different phosphorylation sites in determining arrestin affinity and activation. In addition, all arrestin isoforms bind well to phosphorylated rhodopsin, which is not the case for other GPCRs, so that a direct comparison between the different arrestin isoforms could be performed. For the fluorescence experiments involving the rhodopsin core, we had the additional advantage that significant quantities of rhodopsin in native membranes could be acquired from bovine retinal tissue. Finally, we thought this a logical approach, since most major GPCR discoveries have been first made using rhodopsin. In conclusion, we intend to continue this course of investigation using the C-termini sequences from more GPCRs. A comparison of the different GPCR phosphorylation patterns will certainly be illuminating.

Specific Comments:

1. The method for synthesizing the combinatorial peptide library is very interesting.

Thank you, we agree.

2. Table S1 is a very comprehensive summary of results obtained with the different phosphopeptides, and is referenced heavily in the manuscript. The authors might consider making this table a main figure.

We moved Table S1 to the main text as Table 2.

3. Fig 1b doesn't appear intuitive; an alternate to this black monochrome would be to use a rainbow (or other single-color) heat map for the fitting data.

We have lightened the color intensity of the graph to make grayscale variations more easily distinguishable. We have also added an explanation to the figure legend that clarifies how the heatmap should be interpreted:

"The plot shows the importance (fitting coefficients) of each phosphorylation site for the top 20 models (y-axis) in grayscale. The strength of the coefficient for a site in a model is indicated by the darkness of the boxes. White denotes that the coefficient was set to zero. The darker and longer columns indicate phosphorylation sites important for a tight interaction with arrestin-1. Interception represents the 'non-specific' binding component."

4. Figure 2c needs proper statistical analysis.

We have carried out the statistical analysis as requested. Due to the complexity of presenting this analysis on the graph (3 different arrestin isoforms compared to one another for binding each type of phosphopeptide, as well as many phosphopeptides compared to one another for binding to each arrestin isoform), we have included this information as Supplementary Information Tables S1, S2, and S3.

5. Figure 2d needs some statistics and the parameters (K_d , B_{max}) can be shown as an inset in the same figure

The variance between repeated measurements was small. Therefore we showed three independent experiments instead of the SD values of the individual titration points. We also now indicate B_{max} and K_d in Figure 2d.

6. While figure 4e shows the errors bars as \pm s.e.m, it is unclear if any statistical comparisons are being made. It would be better to indicate (by symbols) the statistical differences for key phosphopeptides.

We have updated the figure panel to show statistical differences (or the absence of them) between individual peptides, for both fluorescence and centrifugal pull-down results. We have included the results of the ANOVA analysis in the figure and an explanation in the legend.

7. While discussing the concept of ‘phosphorylation barcode’, the authors need to discuss the implications of their findings in the context of arrestin function (ex- receptor desensitization, endocytosis/trafficking)

We have added text to the end of the first paragraph of the discussion to put our results into the larger context of arrestin function(s) in the cell:

“To summarize, the GPCR phosphorylation motifs we identify here control not only the overall conformation of arrestin, but its activation state, its ability to couple to the active receptor core, and the relative stability and expected lifetime of the arrestin-GPCR complex. Such knowledge is ultimately necessary to understand the molecular mechanism by which different receptor phosphorylation patterns determine how arrestin interacts with different GPCRs and the resulting cellular functions of arrestin (e.g. receptor desensitization, endocytosis and trafficking).”

8. Authors should show more raw results for the tryptic digests in the supplementary pages. On page 6, the authors discussed the accelerated digestion of arrestin-2 in the presence of 7P, 6P, and 4P peptides but not the 5P peptide. No data was shown for these and the readers are directed to Table S1, which is only a summary.

We have added a new Supplementary Figure S1 to the revised manuscript that shows how the rate of arrestin-2 digestion increased in presence of higher phosphorylated peptides (4< phosphorylated sites), but not in presence of 5P. In addition, we have added text to p.6/7 to better explain the different rates of digestion of arrestin-2:

“Note that digestion of arrestin-2 was accelerated in the presence of 7P, 6P and 4P peptides, but not in the presence of the other binding peptides, including 5P (Table 2,

Supplementary Figure S1). Considering that all peptides containing phosphorylation at both key sites - as well as peptide 5P - bound arrestin-2 with similar high affinity (Table 2), these differences in trypsin sensitivity do not reflect different levels of phosphopeptide association with arrestin-2 but rather differences in the arrestin-2 structure induced by the different peptides."

9. The TROSY-2D NMR data in relation to arrestin conformational changes when bound to various synthetic phosphopeptides is extensive and well done. However, the text related to this needs to be described in a simpler fashion for a general readership. For example, what is being measured spectroscopically (chemical shift/environment) as it relates to conformational changes in arrestin? Also, figures in the paper pertaining to this are not trivial to understand. Perhaps it would be easier to annotate where within arrestin these residues lie, and show this alongside the peaks? Control spectral peaks and peaks attributed to arrestin treated with phosphopeptides should be clearly labeled. It would also be clarifying if the authors annotate the multi-state arrestin peaks (Figure 5, panels a and b in particular). This data is overall very interesting, and the authors are encouraged to discuss them more in the paper.

Regarding the first point, we have added the following sentence to the introduction of the subsection: "*Chemical shift changes of the detected amide signals report changes in the local environment of investigated amino acids, such as changes in polypeptide backbone geometry, H-bonding or polar interactions or changes in its position with respect to charged amino acids and aromatic groups that influences the magnetic field of the studied nuclei.*"

Regarding the annotation of figures with respect to residue location in arrestin, we have already shown this clearly in Figures 6, 7 and SI Figures S4 and S5. Note that Figure 5 of the main text does not require such annotation, since the point of this figure is to show examples of chemical shifts, how they depend on phosphopeptide concentration, and the overall quality of the data. We have added a pictorial legend to Figure 5 to better link the colours of the spectra to the peptide:arrestin ratio in the experiment.

Regarding the annotation of multi-state arrestin peaks in Figure 5, we believe this is not necessary here. Chemical shift changes are already summarized in Figures 6, 7, S4, S5, and SI Table S4. As discussed above, Figure 5 serves as an illustration of observed chemical shifts in arrestin-1 and not an exhaustive summary of important chemical shifts with respect to the arrestin structure.

10. On page 8, the authors stated: "We used [¹⁵N,¹H] TROSY NMR spectroscopy to characterize the location of conformational changes in arrestin-1 induced by differently phosphorylated phosphopeptides (Supplementary Figure S2)." However, the result in Supplementary Figure S2 was obtained with the fully phosphorylated peptide (7P) only and not others. The authors should clarify this in the manuscript.

We have removed the figure reference from the cited sentence, since this sentence serves as an introduction to this subsection of the Results. In the revised manuscript, we have added a sentence a few lines later: "*A representative example of such an NMR experiment to monitor phosphopeptide 7P binding to arrestin-1 is shown in Supplementary Information Figure S3.*

" Note that the revised SI Figure S3 was previously SI Figure S2.

11. In the second paragraph on page 8, Figure 5, 6 and 7 as well as Supplemental Figures S2, S3, S4 are not sufficiently described. Each of these figures should be explained more clearly as to why the authors prepared them and what they are showing.

We have added more text to explain how the data shown were interpreted, so that the non-expert reader can interpret the results.

For example, in the revised legend to Fig. 6, we have added an explanation for the color-code used: *“The rainbow spectrum represents the range of chemical shift perturbations: non-significant (blue), minor (green), moderate (yellow), large (orange), and very large (red). Black marks residues whose signals showed strong line broadening which prevented the final position to be determined, indicating significant changes in mobility of the residues or its surroundings which bring them to the μm -ms range.”*

We also added additional text on p.8/9, to explain the interpretation of the split peaks shown in Figure 5a at high concentrations of phosphopeptide.

12. In figure S1, where the centrifugal pull-down data is shown, the co-binding of light-activated rhodopsin with the phosphopeptide pre-activated arrestin appears to also be mediated by the receptor core binding of arrestin as opposed by solely the phosphorylated receptor tail. The authors should discuss the implications of these data in greater detail.

Receptor core binding by phosphopeptide-bound and -activated arrestin-1 is implicit in the pull-down results, since the rhodopsin is bound within the rod outer segment membranes that are pelleted by centrifugation. Core engagement is also shown by the arrestin-1 I72B mutant, as the fluorescence of this mutant reports the burying of the finger loop within the cytoplasmic crevice of the active receptor (Sommer et al. 2012 Nat Comms). We discuss this fact on p. 7 of the manuscript. In response to this comment, we have added text to the end of the ternary complex results section (p. 8) to further clarify the role of the receptor core in ternary complex formation:

“Notably, the relative levels of ternary complex formation are highly consistent with the other data presented in this study. These data further show that the phosphopeptides that bind arrestin-1 with high affinity and stimulate C-tail release also activate arrestin for receptor core binding. This finding supports the recently described allosteric activation mechanism for arrestin by Latorraca et al. (Latorraca, Wang et al. 2018), where engagement of the binding site for the phosphorylated receptor C-terminus stabilizes an active conformation of arrestin that can couple to the active receptor core.”

13. In figure 7, the author should annotate the structural region (i.e. finger loop, middle loop, polar core, three-element interaction, etc...) of each arrestin residue under investigation as appropriate. This will tune the reader into what important regions are undergoing change upon phosphopeptide binding.

We have now labelled important structural regions and loops in Figure 7 and added a description to the legend.

14. On page 10 of the manuscript, the authors state that pS362 of the V2Rpp (analogous to the inhibitory site pT342 on the rhodopsin tail) in the V2Rpp- β arr1-Fab30 crystal structure is “directed toward the solvent and does not interact with arrestin (Figure 9a).” While the authors are correct that pS362 does appear to be more solvent accessible than other phosphorylated residues on the V2Rpp, pS362 in fact does form an electrostatic interaction with R7 on β arr1. The authors should readdress this point in the manuscript. As far as we are aware, R7 has not been reported to be involved in binding the phosphorylated receptor C-terminus or arrestin activation. In contrast, the key site pS363 interacts with K10 (important phosphosensor) and K107 (part of 3-element interaction), and key site pT360 interacts with the gate loop (part of polar core). R7, however, is not conserved in arrestin-1 (valine is present in analogous location), suggesting the arginine at position 7 in arrestin-2 is not required for arrestin activation to the same extent as the other lysines in the 3-element interaction and polar core (which are conserved in all arrestin isoforms).

In response to this comment, we have added text to the discussion on p. 12 to explain our reasoning: *“The phosphate group at site pS362 appears to form a hydrogen bond to R7 in arrestin-2. However, we do not believe R7 to be important in phosphorylation recognition, since it is not conserved in arrestin-1 and has never been shown to be a critical phosphosensor in arrestin-2 or arrestin-3 (as far as we are aware).”*

In addition, we note that the presence of R7 might explain why arrestin-2 is not sensitive to ‘inhibitory site’ phosphorylation (p. 13):

“In the case of arrestin-2 no similar inhibitory effect of site pT342 was observed. The increased propensity of C-tail displacement in this arrestin isoform might allow the active conformation to be stabilized by a few specific interactions with phosphate groups on the peptide. Alternatively, R7 in arrestin-2 could stabilize pT342 (see above) and thereby negate its inhibitory effect.”

15. Recent structural evidence, namely the rhodopsin-visual arrestin crystal structure (Zhou et al., Cell 2017), indicate that the acidic residue E341 on the rhodopsin tail interacts with key positively charge residues on visual arrestin. Could the authors comment on this crystallographic observation and how that fits in with the data being presented in the manuscript?

We have added the following text at the end of the next-to-last paragraph to address this interesting observation: *“...the current study was (for practical reasons) limited to rhodopsin-derived peptides, and future experiments using peptides derived from other GPCRs will be required to validate the universality of the proposed motifs as well as uncover other receptor-specific arrestin binding determinants. For example, in the crystal structure of the rhodopsin-arrestin-1 complex (Zhou, He et al. 2017), E341, which is positioned between the two key sites, is observed to interact with functionally important charged residues, namely the phosphosensor K15 (Vishnivetskiy, Schubert et al. 2000). This interaction could contribute to the specificity of rhodopsin for arrestin-1 binding (Gurevich, Dion et al. 1995).”*

16. Some statements on page 11, in the discussion section, were made under too many assumptions or are not clear enough:

- “Given the high sequence and structure homology of arrestin isoforms, similarities in activation mechanism by different GPCRs are not entirely unexpected. However certain functional differences are apparent in the presented data.” Here, it is not clear what the authors mean “certain functional differences”.

We have changed the wording here to clarify what we mean (now p. 14):

“Given the high sequence and structure homology of arrestin isoforms, similarities in activation mechanism by different GPCRs are expected. Both the receptor transmembrane helical core and phosphorylated receptor C-terminus contribute to arrestin activation (Gurevich and Benovic 1993, Latorraca, Wang et al. 2018). However, our data suggest that the relative contributions of these two receptor components to activation are different for arrestin-2 and arrestin-3.”

- “Due to its flexible, pre-activated conformation, arrestin-2 robustly binds phosphorylated receptor C-termini, as long as the key phosphorylation sites are present. This preference would explain why arrestin-2 is a relatively poor binder of class A GPCRs²⁸, whose C-termini generally lack the proper spacing between phosphorylation sites to fulfill the requirement to include both the key sites and negatively charged region⁴⁷ (Figure 9b).” Here it is not clear what “this preference” means. The authors did not sufficiently explain how the first sentence can be the reason for what is described in the second sentence.

We have rewritten this section, and added references to the data, to better explain our hypothesis regarding why arrestin-2 is a relatively poor binder of Class A GPCRs (p. 14):

“For arrestin-2, the C-tail tends to displace more in the basal state (Figure 3 and Supplementary Figure S1), which allows arrestin-2 to robustly bind phosphorylated receptor C-termini as long as the key phosphorylation sites are present (Figure 2c and Table 2). This attribute would explain why arrestin-2 is a relatively poor binder of class A GPCRs (Oakley, Laporte et al. 2000), whose C-termini generally lack the proper spacing between phosphorylation sites to fulfill the requirement to include both the key sites and negatively charged region (Oakley, Laporte et al. 2001) (Figure 8c).”

- “Class A GPCRs contain many phosphorylation sites that could potentially fulfill the role of the secondary recruiter sites (pT335 and pS338 in rhodopsin), which could underlie the transient nature of class A interactions⁴⁷.” Reference 47 cited does not support what the authors are describing here.

We thank the reviewer for pointing out this misplaced reference. We have replaced it with Oakley et al. JBC v.275(22) 2000, which is the appropriate reference for this statement.

- “Arrestin-3 binds class A GPCRs better than arrestin-2²⁸ even in the absence of receptor phosphorylation¹⁹, indicating that receptor phosphorylation plays a less significant role as compared to the active receptor core in arrestin-3 activation and binding.” While arrestin-2 and -3 binding to the class A GPCR was not compared to each other in the absence of receptor phosphorylation in either reference 28 or 19 cited here, this sentence made it seem like such a comparison was done.

We thank the reviewer for pointing out the misleading language here. We have changed the language to be more clear, and we have added an additional reference in support of the phosphorylation-independent binding of arrestin-3 to GPCRs: *“Arrestin-3 binds class A GPCRs better than arrestin-2(Oakley, Laporte et al. 2000), and arrestin-3 can even bind GPCRs lacking C-termini (Richardson, Balius et al. 2003, Shukla, Violin et al. 2008). These observations indicate that receptor phosphorylation plays a less significant role as compared to the active receptor core in arrestin-3 activation and binding.”*

- “The GPCR phosphorylation motifs we identify here elucidate the “barcode” that controls the cellular functions of arrestins, by determining both the relative stability of the arrestin-receptor complex and modulating the conformation of the arrestin signaling state.” Unless the authors plan to show that the functional consequences of arrestin are actually governed by the phosphorylation motif identified here, the authors should tone down this sentence.

We have replaced this sentence with the following (p. 12):

“To summarize, the GPCR phosphorylation motifs we identify here control not only the overall conformation of arrestin, but its activation state, its ability to couple to the active receptor core, and the relative stability and expected lifetime of the arrestin-GPCR complex. Such knowledge is ultimately necessary to understand the molecular mechanism by which different receptor phosphorylation patterns determine how arrestin interacts with different GPCRs and the resulting cellular functions of arrestin (e.g. receptor desensitization, endocytosis and trafficking).”

17. It would be a lot more compelling if the authors can show, in a cellular context, that the in vitro results obtained here (i.e. two key sites and an inhibitory site on a phosphorylated GPCR tail) can translate to both rhodopsin and another receptor such as the prototypical class B V2R, using mutagenesis and assessing arrestin recruitment to said receptor via arrestin recruitment assays.

As we describe at length at the beginning of the response to this reviewer, the goal of this study was to define the ‘molecular function’ of each phosphorylation site, as in how each site contributes to affinity, arrestin activation, and arrestin global conformation. We believe the significant insights gained by our in vitro approach warrant publication as they are. Following up with experiments in living cells is beyond the scope of this study and will be addressed in future studies by us and others.

Referencing

In general, the referencing of the manuscript is appropriate and balanced. Two exceptions:

1. The idea of a barcode to explain how GPCRs coordinate the many functions of arrestins was actually proposed about 15 years ago. The appropriate citation is: Kim et al: PNAS 102: 1442-1447, 2005 (<http://www.pnas.org/content/pnas/102/5/1442.full.pdf>)

In the Discussion of this paper the authors propose the barcode hypothesis as follows: “It can be speculated that beta-arrestins bound to receptors phosphorylated on distinctive patterns of sites adopt specific conformations capable of activating different signaling effectors. The location of

these phosphorylation sites may constitute a “bar code” that instructs the bound beta-arrestin as to its intended function. Although details of the preferred phosphorylation consensus motifs for the different GRKs are not well defined, it is clear that there are significant differences in the preferences of the various enzymes (16). Given the large number of potential phosphorylation sites in the cytoplasmic tails of many 7TM receptors, there exists the possibility for combinatorial complexity in the patterns of phosphorylation that might lead to many different conformations of the bound beta-arrestins. The detailed mapping of these sites and correlation with specific functional outcomes should be a profitable direction for future research.”

We thank the reviewer for pointing out this seminal study. We have now included it after the following sentence in the introduction: *“Other studies have suggested that different phosphorylation patterns on the intracellular C-terminal tail (the ‘phosphorylation barcode’) of GPCRs can induce conformationally distinct active states of arrestins that result in a variety of cellular outcomes.”*

2. The authors cite two papers (refs 7 and 8) supporting the view that arrestins “possibly” stimulate G protein independent signaling. In fact, many dozens of papers support the idea that arrestins stimulate both G protein-independent and dependent modes of signaling. And the conclusions of refs 7 and 8, which were performed with Crispr knockout cells, were recently challenged by another study which should be cited together with them (Luttrell L. et al. *Sci Signal*. 2018 Sep 25;11(549). pii: eaat7650. doi: 10.1126/scisignal.aat7650.)

The word ‘possibly’ was intended to acknowledge the ongoing debate in the field regarding the actual role of arrestin in intracellular signalling, and we aimed for a fair and balanced citing of the literature. In line with this approach, we have now included the suggested reference in the manuscript (p.3).

References:

- Azevedo, A. W., T. Doan, H. Moaven, I. Sokal, F. Baameur, S. A. Vishnivetskiy, K. T. Homan, J. J. Tesmer, V. V. Gurevich, J. Chen and F. Rieke (2015). "C-terminal threonines and serines play distinct roles in the desensitization of rhodopsin, a G protein-coupled receptor." *Elife* **4**.
- Gurevich, V. V. and J. L. Benovic (1993). "Visual arrestin interaction with rhodopsin. Sequential multisite binding ensures strict selectivity toward light-activated phosphorylated rhodopsin." *J Biol Chem* **268**(16): 11628-11638.
- Gurevich, V. V., S. B. Dion, J. J. Onorato, J. Ptasienski, C. M. Kim, R. Sterne-Marr, M. M. Hosey and J. L. Benovic (1995). "Arrestin interactions with G protein-coupled receptors. Direct binding studies of wild type and mutant arrestins with rhodopsin, beta 2-adrenergic, and m2 muscarinic cholinergic receptors." *J Biol Chem* **270**(2): 720-731.
- Latorraca, N. R., J. K. Wang, B. Bauer, R. J. L. Townshend, S. A. Hollingsworth, J. E. Olivieri, H. E. Xu, M. E. Sommer and R. O. Dror (2018). "Molecular mechanism of GPCR-mediated arrestin activation." *Nature* **557**(7705): 452-456.
- Mendez, A., M. E. Burns, A. Roca, J. Lem, L. W. Wu, M. I. Simon, D. A. Baylor and J. Chen (2000). "Rapid and reproducible deactivation of rhodopsin requires multiple phosphorylation sites." *Neuron* **28**(1): 153-164.

Oakley, R. H., S. A. Laporte, J. A. Holt, L. S. Barak and M. G. Caron (2001). "Molecular determinants underlying the formation of stable intracellular G protein-coupled receptor-beta-arrestin complexes after receptor endocytosis*." J Biol Chem **276**(22): 19452-19460.

Oakley, R. H., S. A. Laporte, J. A. Holt, M. G. Caron and L. S. Barak (2000). "Differential affinities of visual arrestin, beta arrestin1, and beta arrestin2 for G protein-coupled receptors delineate two major classes of receptors." J Biol Chem **275**(22): 17201-17210.

Richardson, M. D., A. M. Balius, K. Yamaguchi, E. R. Freilich, L. S. Barak and M. M. Kwatra (2003). "Human substance P receptor lacking the C-terminal domain remains competent to desensitize and internalize." J Neurochem **84**(4): 854-863.

Shukla, A. K., J. D. Violin, E. J. Whalen, D. Gesty-Palmer, S. K. Shenoy and R. J. Lefkowitz (2008). "Distinct conformational changes in beta-arrestin report biased agonism at seven-transmembrane receptors." Proceedings of the National Academy of Sciences of the United States of America **105**(29): 9988-9993.

Vishnivetskiy, S. A., C. Schubert, G. C. Climaco, Y. V. Gurevich, M. G. Velez and V. V. Gurevich (2000). "An additional phosphate-binding element in arrestin molecule. Implications for the mechanism of arrestin activation." J Biol Chem **275**(52): 41049-41057.

Zhou, X. E., Y. He, P. W. de Waal, X. Gao, Y. Kang, N. Van Eps, Y. Yin, K. Pal, D. Goswami, T. A. White, A. Barty, N. R. Latorraca, H. N. Chapman, W. L. Hubbell, R. O. Dror, R. C. Stevens, V. Cherezov, V. V. Gurevich, P. R. Griffin, O. P. Ernst, K. Melcher and H. E. Xu (2017). "Identification of Phosphorylation Codes for Arrestin Recruitment by G Protein-Coupled Receptors." Cell **170**(3): 457-469 e413.

Second ROUND

Reviewers' comments:

Reviewer #1 (Remarks to the Author):

The authors tested the role of rhodopsin phosphorylation sites in arrestin-1 binding using an array of differentially phosphorylated peptides derived from the C-terminus of bovine rhodopsin. They measured the affinities of these peptides for visual arrestin-1, as well as non-visual arrestin-2 and arrestin-3, as well as their ability to induce conformational changes in arrestin-1 and to facilitate arrestin-1 binding to unphosphorylated light-activated rhodopsin. The study is very comprehensive, the data are pretty strong and quantitative. Classification of the phosphorylation sites by their functional significance for arrestin binding is novel and thought-provoking. The manuscript was greatly improved in revision, concerns by all reviewers were satisfactorily addressed. There is only one relatively serious deficiency, the idea that there are mechanisms of rhodopsin inactivation independent of its phosphorylation and arrestin binding, which contradicts published findings reproduced by several labs independently. Minor editing is also needed.

1. P. 11. The elimination of either arrestin-1 (Nature. 1997 Oct 2;389(6650):505-9) or GRK1 (rhodopsin kinase) (Proc Natl Acad Sci U S A. 1999 Mar 30;96(7):3718-22) resulted in prolonged rhodopsin signaling, suggesting that phosphorylation- and arrestin-1-independent mechanisms of inactivating rhodopsin in photoreceptors either do not exist, or do not play an

important role. Thus, this authors' hypothesis does not seem plausible. This statement needs to be revised.

We agree with the reviewer that all evidence indicates that deactivation of wild-type rhodopsin is kinase- and arrestin-dependent. However, in the case of threonine-only rhodopsin, which was not effectively phosphorylated, genetic reduction of GRK1 or arrestin did not affect timely shut-off (see Fig. 6 of Azevedo et al. eLife 2015). Hence we conclude that other mechanisms - independent of kinase and arrestin - must be controlling the shut-off of threonine-only rhodopsin.

In the revised manuscript, we have changed the language to specifically refer to this mutant as being kinase- and arrestin-independent.

"These results suggest that other, arrestin-1-independent mechanisms might exist in the rod cell to deactivate this mutant rhodopsin."

2. Minor editing is still needed.

What does the string of numbers on page 6 in "putative phosphopeptide binding site(23604254)" mean?

The reference wasn't displayed correctly in the last version. We changed it and it is displayed correctly now.

P. 9., correct "This is may be"

We changed it to: "*This ability may arise because...*"

p. 11. What is the difference between receptor endocytosis and trafficking?

We removed trafficking.

Reviewer #3 (Remarks to the Author):

None